

# Exact effective action for the O(N) vector model in the large N limit

**Han Ma[1] and Sung-Sik Lee[1,2]**

**1** Perimeter Institute for Theoretical Physics, Waterloo, Ontario N2L 2Y5, Canada
**2** Department of Physics & Astronomy, McMaster University,
1280 Main St. W., Hamilton, Ontario L8S 4M1, Canada

## Abstract

We present the Wilsonian effective action as a solution of the exact RG equation for the critical $O(N)$ vector model in the large $N$ limit. Below four dimensions, the exact effective action can be expressed in a closed form as a transcendental function of two leading scaling operators with infinitely many derivatives. From the exact solution that describes the RG flow from a UV theory to the fixed point theory in the IR, we obtain the mapping between UV operators and IR scaling operators. It is shown that IR scaling operators are given by sums of infinitely many UV operators with infinitely many derivatives.

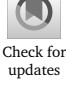

# 1   Introduction

The renormalization group (RG) equation describes the change of effective theories as the energy cutoff is lowered [1–3]. In particular, the infrared (IR) fixed points of the RG equations embody the notion of universality classes, and play the central role in the classification of phases of matter and our understanding of critical phenomena [4–11]. In condensed matter physics, it is also important to understand the RG flow from microscopic theories usually defined on lattices to continuum theories at long distance scales. However, understanding the exact RG flow of effective actions is generally hard for interacting theories. The challenge lies in the fact that operators composed of arbitrary numbers of fields and space derivatives are generated under the exact RG flow.

If a theory comes with a small parameter such as $1/N$ with $N$ being the number of fields, a great deal of simplification arises. The best understood example is the $O(N)$ vector model in the large $N$ limit. In this theory, the exact RG equation for the quantum effective potential for spacetime-independent fields can be written in a closed form thanks to the local potential approximation that becomes exact in the large $N$ limit [3, 12–14]. It allows for a systematic computation of the effective potential in the power series of the field [12, 15–25]. However, the full scale dependent solution of the exact RG equation has not been obtained yet. At the IR fixed point of the exact RG equation, the formal expression of the quantum effective action has been written down as a functional of one collective variable being the O(N) singlet [20, 26]. However, the full relation between the collective field and fundamental microscopic field in the O(N) vector representation is still unknown.

In this paper, we provide additional insights into the exact effective action of the $O(N)$ vector model. The new results of our work are three-folded. Firstly, we obtain the exact solution of the RG equation in the large $N$ limit in terms of the collective variable. The IR limit of our scale dependent effective action is related to the effective action [20, 26] through the Legendre transformation. Then, the saddle point equation allows us to express the collective field in terms of the fundamental fields. Secondly, the full effective action is obtained not

only at the IR fixed point but at general scales. This finds us the exact mapping between UV operators and IR scaling operators. Here, UV operators refer to operators that can be added to a UV Lagrangian while IR operators are the scaling operators that arise in the long-distance limit of the theory, meaning that they are the ones whose form do not change under the coarse graining and dilatation. We note that the relation between UV operators and IR scaling operators is highly non-trivial: turning on a local operator in a lattice model generally amounts to turning on a linear superposition of many IR scaling operators in the continuum theory and conversely, an IR scaling operator corresponds to a sum of multiple UV operators that involve infinitely many derivatives. Based on the exact mapping derived from the solution of the RG equation, we explicitly compute two leading IR scaling operators in terms of the UV fields. As expected, the IR scaling operators involve arbitrarily many fundamental fields and derivatives. Finally, we go beyond the large $N$ limit to compute the leading $1/N$ corrections to the effective action by including fluctuations of the collective field.

To achieve these goals, we use a recently developed quantum RG scheme [27–29], which allows us to keep track of the RG flow starting from a UV theory to the O(N) Wilson Fisher fixed point[1]. As an exact reformulation of the Wilsonian RG, the quantum RG significantly reduces the number of operators that need to be included along the exact RG flow as couplings are promoted to dynamical variables. The information of the operators that are not explicitly kept within the RG flow is encoded in fluctuations of the dynamical couplings. We note that most of the results obtained in our paper for the $O(N)$ vector model through the quantum RG can be in principle obtained through alternative methods that use collective fields. However, the quantum RG can be readily extended to more complicated theories such as matrix models [27,30] for which the method of the collective field used for the vector model is not applicable.

The outline of this paper is as follows. In Sec. 2, we apply the quantum RG method to the O(N) vector model. Subsequently, Sec. 3 gives us the effective action in the IR. In the large N limit, we are able to express this action in terms of the UV field and more surprisingly it can be written in a closed form in terms of scaling operators as discussed in Sec. 4. Furthermore, in Sec. 5, we obtain the generating function from the effective action which allows us to compute correlation functions. Sec. 7 summarizes our main results.

## 2 Quantum renormalization group

Consider the partition function of the $O(N)$ vector model defined on a $D$-dimensional Euclidean lattice,

$$Z = \int \mathcal{D}\phi \, e^{-\frac{m^2}{2}\left[\sum_i \phi_i^2 + \sum_{ij} M_{ij}\phi_i\phi_j\right] - \frac{\lambda}{N}\sum_i (\phi_i^2)^2}, \tag{1}$$

where $\phi_i^a$ is the fundamental field with flavour $a = 1, 2, .., N$ defined at site $i$, $\phi_i\phi_j \equiv \sum_a \phi_i^a \phi_j^a$, $m$ is the on-site mass, $-\frac{m^2}{2}M_{ij}$ is the hopping amplitude between site $i$ and $j$, and $\lambda$ is the on-site quartic interaction. In describing the RG flow, it is convenient to divide the action into a reference action and a deformation added to the reference action. Here we choose the insulating fixed point action, $S_{ref} = \frac{m^2}{2}\sum_i \phi_i^2$ as the reference action. The hopping (kinetic) term and the quartic interaction are regarded as deformations, $S_1 = \frac{\lambda}{N}\sum_i (\phi_i^2)^2 + \frac{m^2}{2}\sum_{ij} M_{ij}\phi_i\phi_j$, which is not necessarily small. Depending on its magnitude, the theory can flow to one of the fixed points in the long-distance limit. Above two dimensions, the possible fixed points include

---

[1]The Wilsonian effective action that satisfies the exact Polchinski RG equation [2] is related to the 1PI effective action through the Legendre transformation.

the insulator, the critical point, and the long-range ordered state. Choosing a reference theory is equivalent to picking a point in the space of theories as the origin. Physics does not depend on this choice[2].

In quantum RG, every local action is associated with a short-ranged entangled quantum state [29]. For the reference action and the deformation, we introduce $|S_{ref}\rangle = \int \mathcal{D}\phi\ e^{-S_{ref}}|\phi\rangle$ and $|S_1\rangle = \int \mathcal{D}\phi\ e^{-S_1}|\phi\rangle$, respectively, where $|\phi\rangle$ is the basis state defined in the $D$ dimensional lattice with the inner product $\langle\phi'|\phi\rangle = \prod_{i,a}\delta(\phi_i'^a - \phi_i^a)$. These are $D$-dimensional states whose wavefunctions are given by the exponentials of the actions. Then the partition function is an overlap between the two states,

$$Z = \langle S_{ref}|S_1\rangle. \tag{2}$$

The renormalization group transformation can be described as a quantum evolution of states associated with actions. An infinitesimal RG transformation is implemented by a quantum evolution operator $e^{-dz\hat{H}}$, where $\hat{H}$ is an RG Hamiltonian and $dz$ is the infinitesimal RG time. As will be shown later, $z$ corresponds to the logarithmic length scale. Since we have chosen the ultra-local action as the reference theory, it is natural to use a coarse graining transformation under which the insulating fixed point action is invariant. The RG Hamiltonian that leaves the insulating fixed point invariant is given by

$$\hat{H} = \sum_i \left[\frac{1}{m^2}\hat{\pi}_i^2 + i\hat{\phi}_i\hat{\pi}_i\right]. \tag{3}$$

$\hat{\pi}_i^a$ is the conjugate momentum of $\hat{\phi}_i^a$ that satisfies the commutation relation, $[\hat{\phi}_i^a, \hat{\pi}_j^b] = i\delta_{ij}\delta_{ab}$. More details about its derivation can be found in Appendix A.1. This RG Hamiltonian is the generator of the exact Polchinski equation [2] in real space [29]. $e^{-dz\hat{\pi}_i^2/m^2}$ has the effect of partially integrating out modes at each site without reducing the number of sites. The remaining 'low-energy' degrees of freedom have less fluctuations and hence a larger mass $me^{dz}$. On the other hand, $e^{-idz\hat{\phi}_i\hat{\pi}_i}$ scales the field as $\phi_i \to e^{-dz}\phi_i$ so that the increased mass in the reference action is put back to the original value. Because $\hat{H}^\dagger|S_{ref}\rangle = 0$, the partition function is invariant under the insertion of the RG evolution operator between the overlap,

$$Z = \langle S_{ref}|e^{-\hat{H}z^*}|S_1\rangle, \tag{4}$$

where $z^*$ increases along the RG flow. Applying the RG evolution operator to $|S_1\rangle$ in Eq. (4), one obtains the state at scale $z^*$, $|S_1^{z^*}\rangle = e^{-\hat{H}z^*}|S_1\rangle$. The state gives the renormalized deformation at scale $z^*$ as $S_1^{z^*} = -\ln\langle\phi|S_1^{z^*}\rangle$ from the state-action correspondence [29]. $S_1^{z^*}$ corresponds to the effective action obtained with the IR cutoff $m^2\frac{e^{-2z^*}}{1-e^{-2z^*}}$ (Appendix A.2). The renormalized action includes infinitely many new higher order interactions, $\sum_{m=2}^\infty J_{i_1,j_1,..;i_m,j_m}(z^*)\frac{1}{N^{m-1}}\prod_{k=1}^m(\phi_{i_k}\phi_{j_k})$, which remain important in the large $N$ limit. In quantum RG, one does not need to keep track of all operators. This simplification arises from the fact that i) the space of theories is viewed as a Hilbert space, and ii) the Hilbert space of $O(N)$ invariant states can be spanned by basis states labeled by the hopping field only,

$$|t\rangle = \int \mathcal{D}\phi\ e^{i\sum_{ij}t_{ij}\phi_i\phi_j}|\phi\rangle. \tag{5}$$

Therefore, $|S_1^z\rangle$ can be expressed as a linear superposition of $|t\rangle$ at all $z$, and one only needs to keep track of the hopping fields along the RG flow. The price to pay is to sum over all RG paths

---

[2]In the appendix, we present the effective action obtained with the choice of the critical Gaussian fixed point theory as the reference action.

for the hopping fields. In other words, the exact Wilsonian RG flow defined in the space of an infinite tower of higher order couplings is expressed as a path integration over the dynamical hopping field only. The $\beta$-functions are then replaced with an action that governs the dynamics of the $z$-dependent hopping field defined in a $(D+1)$-dimensional bulk [27,28,31], where the extra direction corresponds to the length scale $\ln z$. The fact that the $\beta$-functions for all higher order couplings can be encoded in the dynamics of a much smaller subset of couplings also implies that the original $\beta$-functions are highly constrained even in the presence of arbitrary irrelevant deformations [30]. In the phase space path integral representation, $|S_1^{z^*}\rangle$ can be written as

$$|S_1^{z^*}\rangle = \int \mathfrak{D}t\ \mathfrak{D}p\ e^{-NS_{UV}-NS_{bulk}}|t^{z^*}\rangle. \tag{6}$$

Here, $\mathfrak{D}t = \prod_z \mathcal{D}t^z$ and $\mathfrak{D}p = \prod_z \mathcal{D}p^z$ represents the sum over all RG paths. $t_{ij}^z$ is the $z$-dependent dynamical hopping fields between site $i$ and $j$. $p_{ij}^z$ is the conjugate variable that corresponds to the operator $\frac{1}{N}(\phi_i\phi_j)$. As derived in Appendix B, $S_{bulk}$ is the action that determines the weight of each RG path,

$$S_{bulk} = \int_0^{z^*} dz\left[i\sum_{ij}p_{ij}^z\partial_z t_{ij}^z - \frac{2i}{m^2}\sum_k t_{kk}^z + 2i\sum_{kl}t_{kl}^z p_{kl}^z + \frac{4}{m^2}\sum_{kji}t_{ik}^z t_{kj}^z p_{ij}^z\right]. \tag{7}$$

$S_{UV} = \sum_{ij}(it_{ij}^0 + \frac{m^2}{2}M_{ij})p_{ij}^0 + \lambda\sum_i(p_{ii}^0)^2$ is the action for the fields defined at $z=0$. It imposes a dynamical boundary condition for the interacting $O(N)$ model [32,33]. The quadratic term in $p_{ii}^0$ allows $t_{ii}^0$ to have non-trivial fluctuations at the UV boundary. On the contrary, $S_{bulk}$ is linear in $p_{ij}^z$ [34], and $t_{ij}^z$ in the bulk is fixed by $t_{ij}^0$. This fact is a special property of vector models in which the complete basis states are Gaussian in the fundamental field as is shown in Eq. (5) [35–37]. For matrix models, basis states are non-Gaussian, and there exist non-trivial fluctuations of dynamical couplings both at the UV boundary and in the bulk [27]. $S_{bulk}$ in Eq. (7) is related to the one derived in Ref. [29] through a similarity transformation. The bulk theory is finite and well-defined as the original $D$-dimensional field theory is properly regularized [29,31].

## 3  IR deformation in the large N limit

The bulk path integration in Eq. (6) can be readily performed. This allows one to write $|S_1^{z^*}\rangle$ in terms of the integration over $t^0$ and $p^0$ only,

$$|S_1^{z^*}\rangle = \int \mathcal{D}t^0\mathcal{D}p^0 e^{-N\left[S_{UV}-\sum_i\int_0^{z^*}dz\frac{2it_{ii}^z}{m^2}\right]}|t^{z^*}\rangle. \tag{8}$$

Here $t_{ij}^z$ is the solution of $\partial_z t_{ij} + 2t_{ij} - \frac{4i}{m^2}\sum_k t_{ik}t_{kj} = 0$ given by $i\mathbf{t}^z = (i\mathbf{t}^0)\left[e^{2z} - \frac{2}{m^2}(e^{2z}-1)(i\mathbf{t}^0)\right]^{-1}$, where $\mathbf{t}^z$ is a square matrix whose elements are $\{t_{ij}^z\}$ and $\mathbf{t}^0$ is the hopping matrix at $z=0$. The fluctuations of the hopping fields encode higher order operators in Eq. (8). At $z^*=0$, the action for $t^0$ consists of $S_{UV}$ only. Because $S_{UV}$ is Gaussian, integrating over $t^0$ and $p^0$ reproduces the quartic interaction at $UV$. At $z^*>0$, however, the action for $t^0$ becomes non-Gaussian because of the the on-shell bulk action in Eq. (8). The non-Gaussian fluctuation of the hopping fields is what captures higher order couplings in $z^*>0$.

With this picture in mind, we proceed to compute the fixed point action that the theory flows into in the large $z^*$ limit. In the large $N$ limit, the remaining integration can be replaced with the $\phi$-dependent saddle-point. In Eq. (8), $S_{UV}$ is linear in $p_{ij}^0$ for $i \neq j$, and the off-diagonal elements of $t_{ij}^0$ is fixed by the hopping field at UV : $-it_{ij}^0 = \frac{m^2}{2} M_{ij}$ for $i \neq j$. This leaves only the diagonal elements of $t_{ii}^0$ and $p_{ii}^0$ non-trivial. To isolate the deviation of $-it_{ij}^0$ from $\frac{m^2}{2}\mathbf{M}$ for $i = j$, we write $-i\mathbf{t}^0 = \frac{m^2}{2}(\mathbf{M} + \mathbf{X})$, where $\mathbf{X}_{ij} = X_i \delta_{ij}$ is a diagonal matrix. The deformation at $z^*$, $S_1^{z^*} = -\log\langle \phi | S_1^{z^*} \rangle$ becomes

$$S_1^{z^*} = \frac{m^2}{2} \frac{1}{e^{2z^*}-1} \sum_{ij} \left[ I - \frac{e^{2z^*}}{(e^{2z^*}-1)} \frac{1}{\mathbf{K}+\mathbf{X}} \right]_{ij} \phi_i \phi_j + \frac{N}{2} \mathrm{tr}\log\left[(e^{2z^*}-1)(\mathbf{K}+\mathbf{X})\right] - \frac{Nm^4}{16\lambda} \sum_i X_i^2,$$
(9)

up to a constant, where $\mathbf{K} = \left[ \mathbf{M} + \frac{e^{2z^*}}{(e^{2z^*}-1)} I \right]$ with $I$ being the identity matrix, and $\mathbf{X}$ is the saddle point solution satisfying

$$m^2 e^{2z^*} \sum_{kl} (\mathbf{K}+\mathbf{X})_{ik}^{-1}(\phi_k \phi_l)(\mathbf{K}+\mathbf{X})_{li}^{-1} + N(e^{2z^*}-1)^2 \left[ (\mathbf{K}+\mathbf{X})_{ii}^{-1} - \frac{m^4}{4\lambda} X_i \right] = 0, \quad (10)$$

for each and every site $i$ (not summed over). The solution of Eq. (10) can be written in powers of $(\phi_i \phi_j)$ as $X_i(z) = \sum_{m=0}^{\infty} \sum_{j_1,..,j_m} \sum_{k_1,..,k_m} x_{k_1-i,k_2-i,..,k_m-i}^{j_1-i,j_2-i,..,j_m-i}(z) (\phi_{j_1}\phi_{k_1})(\phi_{j_2}\phi_{k_2})..(\phi_{j_m}\phi_{k_m})$, where the rank $2m$ tensor $x_{k_1-i,k_2-i,..,k_m-i}^{j_1-i,j_2-i,..,j_m-i}(z)$ does not depend on $i$ separately because of the translational invariance. The zeroth order coefficient $x(z)$, which is a $z$-dependent function, satisfies the self-consistent equation,

$$\left[ \frac{1}{x(z)I + \mathbf{K}} \right]_{ii} - \frac{m^4}{4\lambda} x(z) = 0. \quad (11)$$

For $D > 2$, $x(z) = x_0 + x_2 e^{-2z^*} + \mathcal{O}(e^{-4z^*})$, where $x_0 \propto \lambda \Lambda^{D-2}$, and $\Lambda$ is the large momentum cutoff at $z^* = 0$ (Appendix D.1).

In the continuum, $\mathbf{M}_{ij}$ can be written as $M(r_i, r_j) = \left[ a - \frac{\nabla_j^2}{m^2} + O(\frac{\nabla_j^4}{m^4}) \right] \delta(r_i - r_j)$, where $a$ is a constant fixed by the UV hopping, and the coefficient of $\nabla^2$ is set to be $-\frac{1}{m^2}$ without loss of generality. The bare mass of the UV theory is given by $m^2(1 + a)$, and $a$ can be used to tune the system across the insulator to symmetry breaking phase transition. The first term in Eq. (9) can be written as $\frac{m^2}{2} \int dr \phi(r) L \phi(r)$, where $L$ is a differential operator which in the large $z^*$ limit takes the following form to the leading order in $\phi$ and the number of derivatives : $L = \frac{e^{2z^*}\left(1+a+x(z^*)-\frac{\nabla^2}{m^2}+O(\frac{\nabla^4}{m^4})+O(\phi^2)\right)}{(e^{2z^*}-1)\left(1+a+x(z^*)-\frac{\nabla^2}{m^2}+O(\frac{\nabla^4}{m^4})+O(\phi^2)\right)+1}$. Next, we would like to discuss different cases characterized by zero or nonzero $\delta$.

## 3.1 Massive theory

For $\delta \equiv 1 + a + x_0 > 0$, $L = 1 + O(e^{-2z})$ in the large $z^*$ limit. In this case, the first term in Eq. (10) is suppressed by $e^{-2z^*}$ compared to the second term at large $z^*$. Consequently, $X_i$ becomes independent of $\phi$, and its saddle-point equation reduces to Eq. (11). In this case, the effective action is simply given by $S_{ref}$. This shows that small hopping and interaction are irrelevant at the ultra-local insulating fixed point. With the strength of hopping increased, the critical point and the long-range ordered state can be reached [29].

### 3.2 Critical theory

At the critical point, the fixed point action becomes qualitatively different. With $\delta = 0$, $\Sigma_{z^*} \equiv e^{2z^*}\big(x(z^*) + a + 1\big) + 1$ approaches $\tilde{\Sigma} \equiv x_2 + 1$ in the large $z^*$ limit. In this case, $L$ is local only at length scale larger than $\frac{e^{z^*}}{\Sigma_{z^*}^{1/2} m}$. This implies that at the critical point the range of the renormalized hopping keeps increasing without a bound with increasing $z$ [29]. In order to have a well-defined large $z^*$ limit at the critical point, one has to scale the coordinate and the field as $\tilde{r} = r e^{-z^*}$, $\tilde{\phi}_{\tilde{r}} = \phi_r e^{\frac{D}{2}z^*}$. This shows that the RG parameter $z$ indeed plays the role of the logarithmic length scale. Accordingly, the rescaling of the effective action and the saddle point equation is carried out in Appendix F. At the critical point, the effective action in Eq. (9) can be written in terms of the scaled variables as

$$\tilde{S}_1^{z^*} = -\frac{1}{2}m^2 \int d^D\tilde{r}\, d^D\tilde{r}' \left[\left(\tilde{\mathbf{T}} + \mathbf{X}'\right)^{-1}_{\tilde{r}\tilde{r}'} \tilde{\phi}_{\tilde{r}} \tilde{\phi}_{\tilde{r}'}\right] + \frac{N}{2}\int d^D\tilde{r}\left[\log\left(\tilde{\mathbf{T}} + \mathbf{X}'\right)\right]_{\tilde{r}\tilde{r}} \qquad (12)$$

$$- N\int d^D\tilde{r}\left\{\frac{m^4}{16\lambda}e^{(D-4)z^*}(X'_{\tilde{r}} - x_2)^2 + \frac{1}{2}\left[\tilde{\mathbf{T}} + x_2 I\right]^{-1}_{\tilde{r}\tilde{r}}(X'_{\tilde{r}} - x_2)\right.$$

$$\left. + \frac{\lambda}{m^4}e^{(4-D)z^*}\left(\left[\tilde{\mathbf{T}} + x_2 I\right]^{-1}_{\tilde{r}\tilde{r}}\right)^2\right\}.$$

Here, $\tilde{\mathbf{T}}_{\tilde{r}\tilde{r}'} = \int^{\Lambda e^{z^*}} \frac{d^D\tilde{Q}}{(2\pi)^D}e^{i\tilde{Q}(\tilde{r}-\tilde{r}')}\left[\frac{\tilde{Q}^2}{m^2} + 1\right]$. $\mathbf{X}'_{\tilde{r}\tilde{r}'} = \delta(\tilde{r} - \tilde{r}')X'_{\tilde{r}}$, where $X'_{\tilde{r}} = e^{2z^*}(X_r - x_0)$. Modulo constant term, the effective action is finite in the large $z^*$ limit[3]. The equation for $\mathbf{X}'$ becomes

$$\frac{m^2}{N}\int d^D\tilde{r}_1 d^D\tilde{r}_2 \left[\tilde{\mathbf{T}} + \mathbf{X}'\right]^{-1}_{\tilde{r}\tilde{r}_1}\tilde{\phi}_{\tilde{r}_1}\tilde{\phi}_{\tilde{r}_2}\left[\tilde{\mathbf{T}} + \mathbf{X}'\right]^{-1}_{\tilde{r}_2\tilde{r}} + \left[\tilde{\mathbf{T}} + \mathbf{X}'\right]^{-1}_{\tilde{r}\tilde{r}} - \left[\tilde{\mathbf{T}} + x_2 I\right]^{-1}_{\tilde{r}\tilde{r}} \qquad (13)$$

$$= \frac{m^4 e^{(D-4)z^*}}{4\lambda}\left(X'_{\tilde{r}} - x_2\right).$$

The $z^*$ dependence of its solution in the large $z^*$ limit is determined by the sign of $D - 4$.

### 3.2.1 $D > 4$

For $D > 4$, the last term in Eq. (13) grows exponentially while other terms remain order one. This forces $X'_{\tilde{r}} = x_2$ in the large $z^*$ limit. In this case, the deformation reduces to a simpler form,

$$\tilde{S}_1^{z^*} = -\frac{m^2}{2}\int d^D\tilde{r}\, \tilde{\phi}_{\tilde{r}}\frac{1}{-\tilde{\nabla}^2/m^2 + \tilde{\Sigma}}\tilde{\phi}_{\tilde{r}} + \mathcal{O}\left(e^{-(D-4)z^*}\tilde{\phi}^4\right). \qquad (14)$$

As $z^*$ increases, the quartic interaction of $\tilde{\phi}$ decays exponentially, and the action approaches the Gaussian form in $\tilde{\phi}$.[4]

---

[3]The second to last term in Eq. (12) is UV divergent in the large $z^*$ limit because $\tilde{\mathbf{T}}_{\tilde{r}\tilde{r}}$ evaluates the matrix element at a coincident point. However, this UV divergence is cancelled by the $\mathbf{X}'$-linear term in $\log\left(\tilde{\mathbf{T}} + \mathbf{X}'\right)$, and the fixed point action is UV finite.

[4]At finite $z^*$, the small but non-zero quartic term generates a mass renormalization. This is why the apparent mass term $(\tilde{\Sigma} - 1)$ does not vanish in general at the critical point. If one starts with the Gaussian theory at UV, $(\tilde{\Sigma} - 1) = 0$.

### 3.2.2  $D < 4$

For $D < 4$, the last term in Eq. (13) can be dropped, and $X'_{\tilde{r}}$ depends on $\tilde{\phi}$. It can be computed order by order in $(\tilde{\phi}_{\tilde{r}}\tilde{\phi}_{\tilde{r}'})$ as $X'_{\tilde{r}} = x_2 + \sum_{k=1}^{\infty} X^{(k)}_{\tilde{r}}$, where $X^{(k)}_{\tilde{r}} \propto (\tilde{\phi}\tilde{\phi})^k$. We can define the Feynman rules as

$$
\begin{aligned}
\frac{m}{\sqrt{N}}\tilde{\phi}^a_{\tilde{r}} &= \bullet \,, \\
(\tilde{\mathbf{T}} + x_2 I)^{-1}_{\tilde{r}\tilde{r}'} &= \tilde{r} \rule[0.5ex]{2em}{0.4pt} \tilde{r}' \,, \\
\left[(\tilde{\mathbf{T}} + x_2 I)^{-1}\right]^2_{\tilde{r}\tilde{r}'} &= \tilde{r} \,\text{\textasciitilde\textasciitilde\textasciitilde}\, \tilde{r}' \,.
\end{aligned}
\tag{15}
$$

The straight line represents the propagator of the free boson field. In the momentum space, it is $G_\phi(p) = p^{-2}$. While the wavy line stands for the propagator of the singlet field which is given by $G_X(p) \propto p^{4-D}$. Diagrammatically, the first and second order terms contributing $X'_{\tilde{r}}$ are given by

$$
\begin{aligned}
X^{(1)}_{\tilde{r}} &= \text{[diagram]} \; \tilde{r} \,, \\
X^{(2)}_{\tilde{r}} &= -2 \,\text{[diagram]}\; \tilde{r} \;+\; \text{[diagram]}\,.
\end{aligned}
\tag{16}
$$

More details are given in Appendix D.2. $X^{(k)}$ is well defined for all $k$ in terms of the scaled variables in the large $z^*$ limit.

We can further expand Eq. (12) in terms of $\tilde{\phi}_{\tilde{r}}$ field. Up to the quartic order, we can get

$$
\frac{1}{N}\tilde{S}^{z^*}_1 = -\frac{1}{2}\int d^D\tilde{r}\,d^D\tilde{r}'(\,\tilde{r} \bullet\!\!\rule[0.5ex]{2em}{0.4pt}\!\!\bullet\tilde{r}'\,) + \frac{1}{4}\int d^D\tilde{r}_1 d^D\tilde{r}_2 d^D\tilde{r}_3 d^D\tilde{r}_4 \left(\text{[diagram]}\right) + \mathcal{O}(\tilde{\phi}^6)\,.
\tag{17}
$$

Higher ordered terms are contributed by all the connected Feynman diagrams with certain numbers of external field $\tilde{\phi}$ represented by black dots. This reproduces the previous result of the effective action as a series of the UV field [14, 20, 25].

## 4  Scaling operators in the large N limit

The exact fixed point effective action in Eq. (12) is a function of $\tilde{\phi}_{\tilde{r}}$ and $X'_{\tilde{r}}$. In order to understand the physical meaning of them, let us deform the hopping amplitude at UV by $-\frac{m^2}{2}\delta M_{ij} = \epsilon\left[\delta_{i,0}\delta_{j,\infty} + \delta_{i,\infty}\delta_{j,0}\right] + \epsilon'\delta_{ij}$. The $\epsilon$-term is an infinite-range hopping, which is equivalent to inserting a pair of fundamental fields : one at the origin and the other at infinity. $\epsilon'$ is a uniform mass deformation. Under these variations, $\mathbf{M}$ and $\mathbf{X}$ are varied. However, only the variation of $\mathbf{M}$ contributes to the change in the renormalized action to the linear order in $\epsilon$ and $\epsilon'$ because the action is stationary with respect to $\mathbf{X}$ at the saddle-point. The variation of the action is studied in Appendix E. In the large $z^*$ limit, it becomes

$$
\delta S^{z^*}_1 = -2\epsilon\,e^{-2\Delta_\phi z^*}\phi^S_0\phi^S_\infty - \epsilon'e^{(D-\Delta_X)z^*}\frac{m^2 N}{2\lambda}\int d^D\tilde{r}\,X'_{\tilde{r}}\,,
\tag{18}
$$

where

$$\phi_{\tilde{r}}^S \equiv \int d\tilde{r}' \left[\tilde{\mathbf{T}} + \mathbf{X}'\right]_{\tilde{r}\tilde{r}'}^{-1} \tilde{\phi}_{\tilde{r}'}, \tag{19}$$

$$X_{\tilde{r}}' = x_2 + \frac{m^2}{N} \frac{\left[\frac{1}{-\tilde{\nabla}^2/m^2+\tilde{\Sigma}}\tilde{\phi}_{\tilde{r}}\right]^2}{\frac{\Gamma(2-\frac{D}{2})}{(4\pi)^{D/2}}\int_0^1 du\left[-u(1-u)\tilde{\nabla}^2/m^2 + \tilde{\Sigma}\right]^{\frac{D}{2}-2}} + \dots \tag{20}$$

$\phi_{\tilde{r}}^S$ is an operator that has the same quantum number as $\phi$, and is made of $\tilde{\phi}$ and $X'$ with smearing at length scale $\tilde{\Sigma}^{-1/2}$ in the scaled variable. $X_{\tilde{r}}'$ is an $O(N)$-singlet composite operator which involves infinitely many $\tilde{\phi}$'s and derivatives (see Appendix D.2 for the expression). Eq. (18) shows that $\phi_{\tilde{r}}^S$ and $X_{\tilde{r}}'$ corresponds to the scaling operators whose forms are invariant under the coarse graining and dilatation. $\phi_{\tilde{r}}^S$ and $X_{\tilde{r}}'$ have scaling dimensions $\Delta_\phi = \frac{D-2}{2}$ and $\Delta_X = 2$, respectively. They are the leading scaling operators in the fundamental and singlet representations of the $O(N)$ group, respectively. We note that Eq. (18) is obtained by taking the large $z^*$ limit of the effective action in the presence of the insertion of two UV operators. At a finite $z^*$, Eq. (18) is corrected by extra terms that are further suppressed by higher powers of $e^{-z^*}$. Since the scaling operators are eigen-operators whose forms do not change under the coarse graining and dilatation, those corrections only contribute to the sub-leading scaling operators with larger scaling dimensions. The expressions for the leading scaling operators in Eqs. (19) and (20) are exact in the large $N$ limit.

The first two terms in Eq. (12) can be written as $\frac{m^2}{2}\int d\tilde{r}d\tilde{r}'\,\phi_{\tilde{r}}^S\left[-(\tilde{\mathbf{T}}+\mathbf{X}')+(\tilde{\mathbf{T}}+\mathbf{X}')^2\right]_{\tilde{r}\tilde{r}'}\phi_{\tilde{r}'}^S$, and the entire fixed point action in Eq. (12) becomes a function of the two scaling operators only. From Eq. (13), one can further find the relation between the two leading scaling operators,

$$\frac{m^2}{N}\phi_{\tilde{r}}^S \times \phi_{\tilde{r}}^S + \left[\tilde{\mathbf{T}} + \mathbf{X}'\right]_{\tilde{r}\tilde{r}}^{-1} - \left[\tilde{\mathbf{T}} + x_2 I\right]_{\tilde{r}\tilde{r}}^{-1} = 0. \tag{21}$$

This gives the exact operator product expansion of two $O(N)$ vector fields in terms of the singlet scaling operator and its descendants.

## 5 Physical observables

The effective action determines all $n$-point functions of the theory in the scaling limit. This follows from the fact that the full generating function,

$$W[J] = -\ln \int \mathcal{D}\phi \; e^{-\frac{m^2}{2}\sum_i \phi_i^2 - S_1[\phi] + \sum_i J_i\phi_i}, \tag{22}$$

can be obtained from the scale dependent generating function

$$W^z[J] = -\ln \int \mathcal{D}\phi \; e^{-\frac{m_z^2}{2}\sum_i \phi_i^2 - S_1[\phi] + \sum_i J_i\phi_i}$$

$$= -\ln \int \mathcal{D}\phi \; e^{-\frac{\sum_i J_i^2}{2m_z^2} - \frac{m_z^2}{2}\sum_i \phi_i^2 - S_1[\phi + J/m_z^2]}, \tag{23}$$

by taking the scaling limit, where $m_z = \frac{m}{\sqrt{1-e^{-2z}}}$. Thus, it is related to the effective action Eq. (12) through

$$W[J] = \lim_{z\to\infty}\left\{-\frac{1}{2m_z^2}\sum_i J_i^2 + S_1[e^z J/m_z^2]\right\}. \tag{24}$$

Then, n-point functions can be obtained by taking derivatives of $W$ with respect to $J$. For example, the 2-point correlation function is given by $G_2^{ab}[r_1, r_2] = \langle \phi_{r_1}^a \phi_{r_2}^b \rangle = -\frac{\delta^2 W}{\delta J_{r_1}^a \delta J_{r_2}^b}$ where $a$ and $b$ are O(N) indices. Using Eq. (9), we can get $G_2^{ab}[r_1, r_2] = \frac{\delta_{ab}}{m^2}[\mathbf{K} + xI]_{r_1,r_2}^{-1}$ where $x$ is the constant part of the saddle point solution of $\mathbf{X}$. Similarly, we can also compute 4-point function of $\phi$ (Appendix G.2) which is given by

$$
G_4^{abcd}[r_1, r_2, r_3, r_4] = \frac{2}{Nm^4} \sum_{rr'} \left( \delta_{ab}\delta_{cd} \;\;\vcenter{\hbox{\includegraphics{x}}}\;\; + \delta_{ac}\delta_{bd} \;\;\vcenter{\hbox{\includegraphics{x}}}\;\; + \delta_{ad}\delta_{bc} \;\;\vcenter{\hbox{\includegraphics{x}}}\;\; \right),
$$

where

$$
\vcenter{\hbox{\includegraphics{x}}} = -(\mathbf{K} + xI)_{r_3,r}^{-1} (\mathbf{K} + xI)_{r,r_4}^{-1} \mathbb{L}_{rr'}^{-1} (\mathbf{K} + xI)_{r',r_1}^{-1} (\mathbf{K} + xI)_{r_2,r'}^{-1} \tag{25}
$$

and $\mathbb{L}$ is a matrix with elements $\mathbb{L}_{rr'} = [(\mathbf{K} + xI)_{rr'}^{-1}]^2 + \frac{m^4}{4\lambda} \delta_{rr'}$. General n-point function can be obtained in the same way.

# 6  1/N corrections

In the large $N$ limit, the integration of the collective variable in Eq. (8) has been evaluated through the saddle-point approximation. For a finite $N$, one has to include fluctuations of the collective fields. Writing fluctuations around the saddle point as $\mathbf{X} = \bar{\mathbf{X}} + \delta\mathbf{X}$ with $\delta\mathbf{X}_{ij} = \delta X_i \delta_{ij}$, we express the effective action as

$$
\begin{aligned}
|S_1^{z^*}\rangle &= \int \mathcal{D}\phi \mathcal{D}\delta X_i \, \exp\Bigg\{ \frac{Nm^4 \sum_i (\bar{X}_i + \delta X_i)^2}{16\lambda} - \frac{N}{2} \text{tr} \log\left[ (1 - e^{-2z^*})(\mathbf{K} + \bar{\mathbf{X}} + \delta\mathbf{X}) \right] \\
&\quad - \frac{m^2}{2(e^{2z^*}-1)} \sum_{ij} \left( I - \frac{e^{2z^*}}{(e^{2z^*}-1)} [\mathbf{K} + \bar{\mathbf{X}} + \delta\mathbf{X}]^{-1} \right)_{ij} \phi_i \phi_j \Bigg\} |\phi\rangle \\
&= \int \mathcal{D}\phi \mathcal{D}\delta X_i \exp\left\{ -N\bar{S}[\phi, \bar{X}] - \Delta S[\phi, \bar{X}, \delta X] \right\} |\phi\rangle,
\end{aligned} \tag{26}
$$

where $\Delta S$ is the action for the fluctuating field,

$$
\Delta S[\phi, \bar{X}, \delta X] = \frac{N}{2} \left[ \sum_{ij} G_{ij}^{-1}[\phi, \bar{X}] \delta X_i \delta X_j + \sum_{ijkl} \Gamma_{ijkl}[\phi, \bar{X}] \delta X_i \delta X_j \delta X_k \delta X_l + \mathcal{O}[(\delta X)^6] \right]. \tag{27}
$$

Here, $G_{ij}[\phi, \bar{X}]$ and $\Gamma_{ijkl}[\phi, \bar{X}]$ are the $\phi$-dependent propagator and quartic vertex for $\delta X$,

$$
\begin{aligned}
G_{ij}^{-1}[\phi, \bar{X}] &= -\frac{m^4}{8\lambda} \delta_{ij} - \frac{1}{2}\left( \frac{e^{2z^*}}{e^{2z^*}-1} [\mathbf{K} + \bar{\mathbf{X}}]_{ij}^{-1} \right)^2 \\
&\quad - \frac{m^2 e^{2z^*}}{2N(e^{2z^*}-1)^2} \sum_{kl} [\mathbf{K} + \bar{\mathbf{X}}]_{ki}^{-1} [\mathbf{K} + \bar{\mathbf{X}}]_{ij}^{-1} [\mathbf{K} + \bar{\mathbf{X}}]_{jl}^{-1} \phi_k \phi_l, \\
\Gamma_{ijkl}[\phi, \delta X] &= -\sum_a [\mathbf{K} + \bar{\mathbf{X}}]_{ai}^{-1} [\mathbf{K} + \bar{\mathbf{X}}]_{ij}^{-1} [\mathbf{K} + \bar{\mathbf{X}}]_{jk}^{-1} [\mathbf{K} + \bar{\mathbf{X}}]_{kl}^{-1} \\
&\quad \times \left[ \frac{1}{4}\left( \frac{e^{2z^*}}{e^{2z^*}-1} \right)^4 \delta_{a,l} + \frac{m^2 e^{2z^*}}{2N(e^{2z^*}-1)^2} \sum_b [\mathbf{K} + \bar{\mathbf{X}}]_{lb}^{-1} \phi_a \phi_b \right]. \tag{28}
\end{aligned}
$$

The first two $1/N$ corrections to the effective action are obtained by integrating over $\delta X$ up to the quartic order in $\delta X$,

$$|S_1^{z^*}\rangle \approx \int \mathcal{D}\phi \exp\left\{-N\bar{S}[\phi]-\delta S[\phi]\right\}|\phi\rangle, \tag{29}$$

where

$$
\begin{aligned}
\delta S[\phi] &= -\log\sqrt{\frac{(2\pi)^{L^2}}{N\det\mathbf{G}^{-1}[\phi,\bar{X}]}}\left[1+\frac{N}{2}\sum_{abcd}\Gamma_{abcd}[\phi,\bar{X}]\langle\delta X_a\delta X_b\delta X_c\delta X_d\rangle\right] \\
&= \frac{1}{2}\left(\log\left[\det\mathbf{G}^{-1}\right]+\mathrm{const}\right)\left[1+\frac{1}{2N}\sum_{abcd}\Gamma_{abcd}\left(G_{ab}G_{cd}+G_{ac}G_{bd}+G_{ad}G_{bc}\right)\right]. \tag{30}
\end{aligned}
$$

Higher order corrections can be similarly obtained.

The fluctuating collective variable $\delta X$ makes the hopping field $t_{ij}$ dynamical in the bulk. Since the emergent geometry that the low-energy field is subject to at scale $z$ is controlled by the hopping field at that scale, the fluctuating hopping field makes the bulk spacetime geometry dynamical [28, 29]. However, the nature of the dynamical spacetime that emerges from Eq. (26) is rather special because the collective field in the bulk fluctuates only through fluctuations of $\delta X$ at $z = 0$ in Eq. (8). For a configuration of the hopping field at the UV boundary, the hopping field in $z > 0$ is completely fixed[5]. This peculiarity arises due to the fact that in the vector model the theory that only includes the single-trace operators is free [28].

There is a way to obtain an alternative bulk theory in which the hopping field exhibit non-trivial fluctuations even in the bulk. This can be achieved by including a quartic interaction in the reference action as [29]

$$
\begin{aligned}
S'_{ref} &= \frac{m^2}{2}\sum_i\phi^2+\frac{\lambda}{N}\sum_i\left(\phi_i^2\right)^2, \\
S'_1 &= \frac{m^2}{2}\sum_{ij}M_{ij}\phi_i\phi_j. \tag{31}
\end{aligned}
$$

Here, the quartic interaction is moved from $S_1$ to $S_{ref}$ and the partition function $Z = \langle S_{ref}|S_1\rangle = \langle S'_{ref}|S'_1\rangle$ is unchanged. The RG Hamiltonian that leaves the new reference action invariant is obtained from Eq. (3) through a similarity transformation,

$$
\begin{aligned}
\hat{H}' &= e^{\frac{\lambda}{N}\sum_i(\phi_i^2)^2}\hat{H}e^{-\frac{\lambda}{N}\sum_i(\phi_i^2)^2} \\
&= \sum_i\left[i\phi_i\pi_i-\frac{4\lambda}{N}(\phi_i^2)^2+\frac{1}{m^2}\pi_i^2+\frac{i}{m^2}\frac{8\lambda}{N}\phi_i^2\phi_i\pi_i+\frac{4\lambda}{m^2}(1+\frac{2}{N})\phi_i^2-\frac{16}{m^2}\frac{\lambda^2}{N^2}(\phi_i^2)^3\right]. \tag{32}
\end{aligned}
$$

Accordingly, the bulk action becomes

$$
\begin{aligned}
S'_{bulk}[z^*] = \int_0^{z^*}dz&\left[i\sum_{kl}p_{kl}^z\partial_z t_{kl}^z-\frac{2i}{m^2}\sum_k t_{kk}^z+2i\sum_{kl}t_{kl}^z p_{kl}^z+\frac{4}{m^2}\sum_{ijk}t_{ik}^z t_{kj}^z p_{ij}^z\right. \\
&\left.+\sum_k\left(\frac{4\lambda(1+\frac{2}{N})}{m^2}p_{kk}^z-\frac{16\lambda^2}{m^2}(p_{kk}^z)^3-4\lambda(p_{kk}^z)^2\right)+\frac{16i\lambda}{m^2}\sum_{kl}t_{kl}^z p_{ll}^z p_{kl}^z\right]. \tag{33}
\end{aligned}
$$

---

[5]Inside the bulk, $p_{ij}$ acts as a Lagrangian multiplier that suppresses the fluctuations of $t_{ij}$ [34].

In this alternative but exactly equivalent formulation, the bulk theory includes the hopping field that is genuinely dynamical. The UV action is now linear in $p_{ij}$ : $S'_{UV} = \sum_{ij}(it^0_{ij} + \frac{m^2}{2}M_{ij})p^0_{ij}$. Integration over $p^0_{ij}$ at the UV boundary imposes the Dirichlet boundary condition for the dynamical source at $z = 0$, $t^0_{ij} = i\frac{m^2}{2}M_{ij}$. The partition function is given by the functional integration of the dynamcal source in the bulk,

$$|S^{z^*}\rangle = \int \mathcal{D}\phi \left[\int \mathfrak{D}t^{z>0}\mathfrak{D}p^{z>0}\, e^{-NS'_{bulk}[t,p]+i\sum_{ij}t_{ij}\phi_i\phi_j}\Big|_{t^0_{ij}=i\frac{m^2}{2}M_{ij}}\right]|\phi\rangle. \tag{34}$$

In the large $N$ limit, this expression is reduced to

$$|S^{z^*}\rangle \approx \int \mathcal{D}\phi\, e^{-NS'_{bulk}[\bar{t},\bar{p},z^*]+i\sum_{ij}\bar{t}^{z^*}_{ij}\phi_i\phi_j}|\phi\rangle, \tag{35}$$

with fields $t$ and $p$ replaced by $\bar{t}$ and $\bar{p}$ as a saddle point solution of equations [29]

$$i\partial_z p^z_{ij} = -\frac{2i}{m^2}\delta_{ij} + 2ip^z_{ij} + \frac{4}{m^2}\sum_k(t^z_{jk}p^z_{ik} + t^z_{ki}p^z_{kj}) + \frac{8i\lambda}{m^2}\left(p^z_{ii} + p^z_{jj}\right)p^z_{ij},$$

$$i\partial_z t^z_{ij} = -2it^z_{ij} - \frac{4}{m^2}\sum_k t^z_{ki}t^z_{kj} - \frac{4\lambda}{m^2}\delta_{ij} + 8\lambda\delta_{ij}\left(\frac{6\lambda}{m^2}p^z_{ii} + 1\right)p^z_{ii}$$

$$-\frac{8i\lambda}{m^2}\delta_{ij}\sum_k(t^z_{ki}p^z_{ki} + t^z_{ik}p^z_{ik}) - \frac{8i\lambda}{m^2}t^z_{ij}\left(p^z_{jj} + p^z_{ii}\right). \tag{36}$$

The $1/N$ corrections can be incorporated by including fluctuations of the collective fields as

$$|S^{z^*}_1\rangle = \int \mathcal{D}\phi\mathfrak{D}\delta t^{z>0}\mathfrak{D}\delta p^{z>0}e^{-NS'_{bulk}[\bar{t}+\delta t,\bar{p}+\delta p,z^*]+i\sum_{ij}\bar{t}^{z^*}_{ij}\phi_i\phi_j}|\phi\rangle, \tag{37}$$

where

$$S'_{bulk}[\bar{t}+\delta t,\bar{p}+\delta p,z^*] \tag{38}$$

$$= S'_{bulk}[\bar{t},\bar{p},z^*] + \int_0^{z^*}dz\Big(i\sum_{kl}\delta p^z_{kl}\partial_z\delta t^z_{kl} + 2i\sum_{kl}\delta t^z_{kl}\delta p^z_{kl} + \frac{4}{m^2}\sum_{ijk}\delta t^z_{ik}\delta t^z_{kj}\bar{p}^z_{ij}$$

$$+ \frac{4}{m^2}\sum_{ijk}(\delta t^z_{ik}t^z_{kj}\delta p^z_{ij} + t^z_{ik}\delta t^z_{kj}\delta p^z_{ij}) - 4\lambda\sum_k\left[\frac{12\lambda}{m^2}\bar{p}^z_{kk} + 1\right](\delta p_{kk})^2$$

$$+ \frac{16i\lambda}{m^2}\sum_{kl}\left[\delta t^z_{kl}\delta p^z_{ll}\bar{p}^z_{kl} + \delta t^z_{kl}\bar{p}^z_{ll}\delta p^z_{kl} + \bar{t}^z_{kl}\delta p^z_{ll}\delta p^z_{kl}\right]\Big) + \mathcal{O}\big[(\delta t)^3, (\delta p)^3\big]. \tag{39}$$

Integration over $\delta t$ and $\delta p$ gives the leading $1/N$ correction to the effective action as in Eq. (30).

# 7 Summary and discussion

The main result of the paper is Eq. (12), which is a closed form of the Wilsonian effective action for the vector O(N) model in the large $N$ limit. Two comments are in order. First, the effective action evaluated at $z^*$ is local at length scales $r \gg e^{z^*}/m$ (equivalently, $\tilde{r} \gg 1/m$). This is because the effective action in Eq. (12) is obtained with the IR cutoff $me^{-z^*}$. The full effective action obtained for $z^* = \infty$ is non-local at the critical point. Second, the form of

the effective action depends on the RG scheme because the way IR cutoff is imposed depends on the scheme (i.e., the choice of $S_{ref}$). Nonetheless, the effective action for the modes with $q \gg m e^{-z^*}$ ($\tilde{q} \gg m$) should be independent of RG scheme. Furthermore, the effective action at large momenta takes non-local form as expected.

In conclusion, we obtained the exact Wilsonian effective action of the interacting $O(N)$ vector model. It takes a closed form of a transcendental function of the two leading scaling operators in the large N limit, where one is in the fundamental representation of $O(N)$ and the other is the singlet. It will of great interest to extend the present result to fermionic systems and theories with non-local/imaginary couplings [38,39].

# Acknowledgements

Research at Perimeter Institute is supported in part by the Government of Canada through the Department of Innovation, Science and Economic Development Canada and by the Province of Ontario through the Ministry of Colleges and Universities. SL acknowledges the support of the Natural Sciences and Engineering Research Council of Canada.

# A  RG scheme

## A.1  RG Hamiltonian

In this section, we review the derivation of the RG Hamiltonian in Eq. (3), and elaborate why the quantum evolution generated by the RG Hamiltonian can be understood as a coarse graining transformation [29]. Consider a theory of a scalar whose partition function is

$$Z = \int \mathcal{D}\phi \; e^{-S_{ref}-S_1}, \tag{A.1}$$

where $S_{ref} = \frac{1}{2}m^2 \sum_i \phi_i^2$ is the reference action chosen to be the insulating fixed point action, and $S_1$ includes the kinetic term and all interactions. As is discussed in the main text, this partition function can be written as the overlap between two quantum states, $Z = \langle S_{ref}|S_1 \rangle$, where $|S_{ref}\rangle = \int \mathcal{D}\phi e^{-S_{ref}}|\phi\rangle$ and $|S_1\rangle = \int \mathcal{D}\phi e^{-S_1[\phi]}|\phi\rangle$.

In the real-space RG à la Kadanoff, a block of sites is merged into a coarse grained site at each step. This forces the RG step to be discrete. To avoid this, we adopt the real space RG scheme that keeps the number of sites unchanged under coarse graining. In this scheme, the field at each site is partially integrated out by an infinitesimal amount without removing the site. To facilitate this, we introduce an auxiliary field $\Phi$ with mass $\mu$ to the action as

$$S' = \frac{1}{2}m^2 \sum_i \phi_i^2 + \frac{1}{2}\sum_i \mu^2 \Phi_i^2 + S_1[\phi]. \tag{A.2}$$

Now the physical field and the auxiliary field are rotated into low-energy mode ($\phi'$) and high-energy mode ($\tilde{\phi}$) as

$$\phi_i = \phi'_i + \tilde{\phi}_i, \qquad \Phi_i = A\phi'_i + B\tilde{\phi}_i, \tag{A.3}$$

where $A = \frac{m^2}{\tilde{\mu}\mu}$ and $B = -\frac{\tilde{\mu}}{\mu}$ with $\tilde{\mu} = \frac{m}{\sqrt{e^{2dz}-1}} \approx \frac{m}{\sqrt{2dz}}$. Through this basis transformation, the low-energy mode acquires a new mass, $me^{dz}$. The increased mass suppresses fluctuations of the low-energy mode slightly. The missing fluctuation has been transferred to the high-energy

mode, which has a heavy mass with order of $m/\sqrt{dz}$. In order to restore the reference action for the low-energy mode, the fields are scaled as

$$\phi_i' = e^{-dz}\phi_i'', \qquad \tilde{\phi}_i = e^{-dz}\tilde{\phi}_i''. \tag{A.4}$$

In the new basis, the action reads

$$S'' = \frac{1}{2}m^2 \sum_i (\phi_i'')^2 + \frac{1}{2}\sum_i \tilde{\mu}^2 (\tilde{\phi}_i'')^2 + S_1[e^{-dz}(\phi_i'' + \tilde{\phi}_i'')]. \tag{A.5}$$

Integrating out the heavy mode $\tilde{\phi}''$, the action is renormalized by $\delta S_1$ which is given by

$$e^{-S_1[\phi_i'']-\delta S_1[\phi_i'']} = \left[ 1 - dz\,\phi_i''\frac{\partial}{\partial\phi_i''} + \frac{dz}{m^2}\left(\frac{\partial}{\partial\phi_i''}\right)^2 \right] e^{-S_1[\phi'']}. \tag{A.6}$$

This is the real space version of the exact Polchinski RG equation [2]. The renormalization of the deformation can be understood as a result from a quantum evolution acting on the wavefunction $e^{-S_1}$ : $e^{-S_1[\phi]-\delta S_1[\phi]} = e^{-\hat{H}dz}e^{-S_1[\phi]}$, where $\hat{H}$ is the RG Hamiltonian,

$$\hat{H} = \sum_i \left[ i\phi_i\pi_i + \frac{1}{m^2}\pi_i^2 \right], \tag{A.7}$$

with $\pi_i = -i\frac{\partial}{\partial\phi_i}$. One can readily check $H^\dagger|S_{ref}\rangle = 0$. It is straightforward to generalize this to the case for the $N$-component scalar fields as is shown in Eq. (3).

## A.2  IR cutoff of the effective action

The renormalized deformation at scale $z$ is given by $S_1^z = -\ln\langle\phi|S_1^z\rangle$, where $|S_1^z\rangle = e^{-\hat{H}z}|S_1^0\rangle$. In this section, we show that this renormalized action indeed represents the effective action obtained with a $z$-dependent IR cutoff. From the discussion in the previous subsection, the renormalized action after one infinitesimal RG step can be written as

$$e^{-S_1^{dz}[\phi^{(1)}]} = \int \mathcal{D}\tilde{\phi}^{(1)} e^{-\frac{\tilde{\mu}^2}{2}\sum_i(\tilde{\phi}_i^{(1)})^2 - S_1[e^{-dz}(\phi^{(1)}+\tilde{\phi}^{(1)})]}. \tag{A.8}$$

After one more step of coarse graining, the renormalized action becomes

$$e^{-S_1^{2dz}[\phi^{(2)}]} = \int \mathcal{D}\tilde{\phi}^{(1)}\mathcal{D}\tilde{\phi}^{(2)} e^{-\frac{\tilde{\mu}^2}{2}\sum_i\left[(\tilde{\phi}_i^{(1)})^2+(\tilde{\phi}_i^{(2)})^2\right]-S_1[e^{-2dz}\phi^{(2)}+e^{-2dz}\tilde{\phi}^{(2)}+e^{-dz}\tilde{\phi}^{(1)}]}. \tag{A.9}$$

Repeating this for $n$ steps, we obtain

$$e^{-S_1^{ndz}[\phi^{(n)}]} = \int \prod_k \mathcal{D}\tilde{\phi}^{(k)} e^{-\frac{\tilde{\mu}^2}{2}\sum_i\sum_{k=1}^n(\tilde{\phi}_i^{(k)})^2 - S_1[e^{-ndz}\phi^{(n)}+\sum_k e^{-kdz}\tilde{\phi}^{(k)}]}. \tag{A.10}$$

Defining $\phi_{i,<} = e^{-ndz}\phi_i^{(n)}$ and $\phi_{i,>} = \sum_k e^{-kdz}\tilde{\phi}_i^{(k)}$ with $n = z^*/dz$, we obtain

$$e^{-S_1^{z^*}[e^{z^*}\phi_<]} = \int \mathcal{D}\phi_> \, e^{-\frac{m_{z^*}^2}{2}\sum_i\phi_{i,>}^2 - S_1[\phi_<+\phi_>]}, \tag{A.11}$$

where $m_{z^*} = \frac{m}{\sqrt{1-e^{-2z^*}}}$. To obtain Eq. (A.11), we insert $1 = \int \mathcal{D}\phi_>\mathcal{D}\gamma\, e^{i\sum_i\gamma_i(\phi_{>,i}-\sum_k e^{-kdz}\tilde{\phi}_i^{(k)})}$ in the integrand of Eq. (A.10), replace $\sum_k e^{-kdz}\tilde{\phi}^{(k)}$ with $\phi_>$ in $S_1$, and integrate over $\tilde{\phi}^{(k)}$ and $\gamma$. We note that Eq. (A.11) is the effective action of a background field $\phi_<$ for a theory whose mass is greater than the mass of the original theory by $\delta m^2 = m^2\frac{e^{-2z}}{1-e^{-2z}}$. If the original theory is at the critical point, the new theory defined at $z^*$ has an IR cutoff $\delta m^2$. The IR cutoff varies from infinity to zero as $z^*$ changes from zero to infinity. The one particle irreducible effective action with the $z$-dependent IR cutoff, which satisfies the exact Wetterich RG equation [3], can be readily obtained from $S_1^z$ through the Legendre transformation [12].

## B  Bulk action

In this section, we derive the bulk action in Eq. (7). We start with $|S_1^0\rangle$ that defines the deformation added to the insulating fixed point action at UV,

$$|S_1^0\rangle = \int \mathcal{D}\phi \, e^{-\left[\frac{\lambda}{N}\sum_i(\phi_i^2)^2 + \frac{m^2}{2}\sum_{ij}M_{ij}\phi_i\phi_j\right]}|\phi\rangle. \tag{B.1}$$

This state can be expanded as a linear superposition of $|t\rangle$ at $z = 0$ defined in Eq. (5) as

$$|S_1^0\rangle = \int \mathcal{D}t_{ij}^0 \int \mathcal{D}p_{ij}^0 \int \mathcal{D}\phi \; e^{-i\sum_{ij}t_{ij}^0(Np_{ij}^0-\phi_i\phi_j)} e^{-\left[\frac{m^2}{2}\sum_{ij}M_{ij}\phi_i\phi_j + \frac{\lambda}{N}\sum_i(\phi_i^2)^2\right]}|\phi\rangle$$

$$= \int \mathcal{D}t_{ij}^0 \int \mathcal{D}p_{ij}^0 \; e^{-NS_{UV}[t_{ij}^0,p_{ij}^0]}|t^0\rangle, \tag{B.2}$$

where $S_{UV}[t_{ij}^0,p_{ij}^0] = \sum_{ij}(it_{ij}^0 + \frac{m^2}{2}M_{ij})p_{ij}^0 + \lambda\sum_i(p_{ii}^0)^2$. The $O(N)$ symmetry guarantees that the wavefunction for $|S_1^z\rangle = e^{-\hat{H}z}|S_1^0\rangle$ is a function of the bi-linear $\phi_i\phi_j$, and $|S_1^z\rangle$ can be also spanned by $|t\rangle$ at $z$ as

$$|S_1^z\rangle = \int \mathcal{D}t_{ij}^z \int \mathcal{D}p_{ij}^z \int \mathcal{D}\phi \; e^{-i\sum_{ij}t_{ij}^z(Np_{ij}^z-\phi_i\phi_j)} e^{-S_1^z[\phi_i\phi_j]}|\phi\rangle$$

$$= \int \mathcal{D}t_{ij}^z \int \mathcal{D}p_{ij}^z \; e^{-N\sum_{ij}it_{ij}^zp_{ij}^z - S_1^z[Np_{ij}^z]}|t^z\rangle, \tag{B.3}$$

where $S_1^z[\phi_i\phi_j] = -\ln\langle\phi|S_1^z\rangle$, and the superscript $z$ for $t$, $p$ and $|t\rangle$ denotes RG time. Therefore, it is enough to understand the evolution of the basis state under the RG Hamiltonian. The evolution from $z$ to $z + dz$ of the basis state is given by

$$e^{-\hat{H}[\hat{\phi},\hat{\pi}]dz}|t^z\rangle$$

$$= \int \mathcal{D}\phi \left[1 - \sum_k\left(\phi_k\frac{\partial}{\partial\phi_k} - \frac{1}{m^2}\frac{\partial^2}{\partial\phi_k^2}\right)dz\right] e^{i\sum_{ij}t_{ij}^z\phi_i\phi_j}|\phi\rangle$$

$$= \int \mathcal{D}\phi \, \exp\left[-2\left(-\frac{iN}{m^2}\sum_k t_{kk}^z + i\sum_{kl}t_{kl}^z\phi_k\phi_l + \frac{2}{m^2}\sum_{kji}t_{ki}^z t_{kj}^z\phi_i\phi_j\right)dz\right] e^{i\sum_{ij}t_{ij}^z\phi_i\phi_j}|\phi\rangle$$

$$= \int \mathcal{D}t_{ij}^{z+dz}\mathcal{D}p_{ij}^{z+dz} e^{-dzN\mathcal{L}_{bulk}}|t^{z+dz}\rangle, \tag{B.4}$$

where $\mathcal{L}_{bulk}$ is the bulk Lagrangian with

$$\mathcal{L}_{bulk} = i\sum_{ij}p_{ij}^{z+dz}\partial_z t_{ij}^{z+dz} - \frac{2i}{m^2}\sum_k t_{kk}^{z+dz} + 2i\sum_{kl}t_{kl}^{z+dz}p_{kl}^{z+dz} + \frac{4}{m^2}\sum_{kji}t_{ki}^{z+dz}t_{kj}^{z+dz}p_{ij}^{z+dz}. \tag{B.5}$$

From this, one can write $|S_1^z\rangle$ as the path integration shown in Eq. (6).

## C  Derivation of the saddle-point equation in Eq. (10)

In Eq. (7), $p_{ij}$ acts as a Lagrange multiplier that enforces the constraint

$$\partial_z t_{ij} + 2t_{ij} - \frac{4i}{m^2}\sum_k t_{ki}t_{kj} = 0 \tag{C.1}$$

in the bulk. We treat $t_{ij}$ as a matrix. The solution $i\mathbf{t}^z[\mathbf{t}^0] = (i\mathbf{t}^0)\left[e^{2z} - \frac{2}{m^2}(e^{2z}-1)(i\mathbf{t}^0)\right]^{-1}$ can be written as a function of $z$ and $\mathbf{t}^0$. The partition function becomes

$$Z = \int \mathcal{D}t_{ij}^0 \mathcal{D}p_{ij}^0 \left\langle S_{ref} | e^{-NS_{UV}[\mathbf{t}^0, \mathbf{p}^0] + N\frac{2i}{m^2}\sum_i \int_0^{z^*} dz\, t_{ii}^z[\mathbf{t}^0]} | t^{z^*}[\mathbf{t}^0] \right\rangle . \tag{C.2}$$

Next, we integrate over $p_{ij}^0$ at the UV boundary to obtain

$$Z = \int \mathcal{D}\phi \int \mathcal{D}t_{ij}^0\, e^{-S_{tot}} \left[ \prod_{i \neq j} \delta\left( t_{ij}^0 - i\frac{m^2}{2}M_{ij} \right) \right], \tag{C.3}$$

where the total effective action is given by

$$S_{tot}[\phi, \mathbf{X}] = \frac{1}{2}m^2 \sum_i \phi_i^2 + \frac{1}{2}m^2 \sum_{ij} \left( (\mathbf{X}+\mathbf{M})\left[ (e^{2z^*}-1)(\mathbf{X}+\mathbf{M}+I) + I \right]^{-1} \right)_{ij} \phi_i \phi_j \tag{C.4}$$

$$+ N\sum_i \int_0^{z^*} dz \left( (\mathbf{X}+\mathbf{M})\left[ (e^{2z}-1)(\mathbf{X}+\mathbf{M}+I) + I \right]^{-1} \right)_{ii} - \frac{Nm^4}{16\lambda} \sum_i X_i^2 .$$

Here, the diagonal element of $t_{ij}^0$ which is not fixed by the delta function is singled out as $-\frac{2i\mathbf{t}^0}{m^2} = \mathbf{X}+\mathbf{M}$ and $\mathbf{X}_{ij} = X_i \delta_{ij}$ is a diagonal matrix. Integrating over $t_{i \neq j}^0$ and $\mathbf{X}$ in the large $z^*$ limit gives the the IR fixed point action. In the large N limit, we can use the saddle point approximation. The variation of the first two terms with respect to $\mathbf{X}$ gives

$$\sum_{jk} \partial_{\mathbf{X}_{ii}} \left[ (\mathbf{X}+\mathbf{M})\left( I + (e^{2z^*}-1)(\mathbf{X}+\mathbf{M}+I) \right)^{-1} \right]_{ij} \phi_i \phi_j$$

$$= e^{2z^*} \sum_{jk} \left[ I + (e^{2z^*}-1)(\mathbf{X}+\mathbf{M}+I) \right]_{ij}^{-1} \phi_j \phi_k \left[ I + (e^{2z^*}-1)(\mathbf{X}+\mathbf{M}+I) \right]_{ki}^{-1} , \tag{C.5}$$

without summation of $i$ index, and the third term gives

$$\partial_{\mathbf{X}} \mathrm{Tr} \log\left[ (e^{2z^*}-1)(\mathbf{X}+\mathbf{M}+I) + 1 \right] = (e^{2z^*}-1)\left[ I + (e^{2z^*}-1)(\mathbf{X}+\mathbf{M}+I) \right]^{-1} . \tag{C.6}$$

Collecting these two contributions, we obtain the saddle-point equation in Eq. (10).

# D    Saddle point solution

In this section, we present the saddle-point solution for $\mathbf{X}$. We will separately discuss $x(z)$, which is the $\phi$ independent part, and $\mathbf{X} - x(z)$ which is $\phi$ dependent.

## D.1    Field independent part

The field independent part of $\mathbf{X}$ denoted as $x$ satisfies Eq. (11),

$$(e^{2z}-1)\left[ (e^{2z}-1)(xI+\mathbf{M}+I) + I \right]_{ii}^{-1} - \frac{m^4}{4\lambda}x = 0 . \tag{D.1}$$

Fourier transformation of the matrix can be implemented through a unitary transformation, and the LHS becomes

$$\left[ (e^{2z}-1)(xI+\mathbf{M}+I) + I \right]^{-1} = U\left[ (e^{2z}-1)(xI+\mathbf{M}'+I) + I \right]^{-1} U^{-1} , \tag{D.2}$$

where $U_{in} = \frac{1}{\sqrt{V}}e^{iQ_n r_i}$, $V$ is the number of sites, and $\mathbf{M}' = U^{-1}\mathbf{M}U$ which is given by $M'_{mn} = \sum_{ij} U^{-1}_{mi} M_{ij} U_{jn} = \frac{1}{V}\sum_{ij} e^{-iQ_m r_i} M_{ij} e^{iQ_n r_j} = M'_{Q_n}\delta_{mn}$. Above, we use the fact that $M_{ij}$ depends only on the separation between $i$ and $j$ in the presence of the translational invariance. In the momentum space, the equation for $x$ becomes

$$\frac{1}{V}\sum_n \frac{1}{x + \frac{e^{2z}}{e^{2z}-1} + M'_{Q_n}} = \frac{m^4}{4\lambda}x\,. \tag{D.3}$$

Here we use the simple dispersion $M'_Q = a + \frac{Q^2}{m^2}$ with a hard momentum cutoff $\Lambda$. For $D > 2$, the left hand side of the equation becomes

$$\int^\Lambda \frac{d^D Q}{(2\pi)^D}\frac{1}{\frac{Q^2}{m^2} + x + \frac{e^{2z}}{e^{2z}-1} + a} = c_1 m^2 \Lambda^{D-2} + c_2 m^4\left(x + \frac{e^{2z}}{e^{2z}-1} + a\right)\Lambda^{D-4} + O\left(\Lambda^{D-6}\right),$$
$$\tag{D.4}$$

where $c_i$ are constants. Writing $x = x_0 + x_2 e^{-2z} + ..$, at the critical point, we obtain

$$x_0 = c_1 \frac{4\lambda}{m^2}\Lambda^{D-2}\,, \qquad x_2 = \frac{c_2\Lambda^{D-4}}{\frac{1}{4\lambda} - c_2\Lambda^{D-4}}\,, \tag{D.5}$$

to the leading order in $\Lambda$.

## D.2 Field dependent part

In $D > 4$, $\mathbf{X}' = x_2$. Here we compute $\mathbf{X}'$ in $D < 4$. At large $z^*$, the saddle point equation in Eq. (13) becomes

$$\int d^D\tilde{r}_1 d^D\tilde{r}_2 \left[\tilde{\mathbf{T}}' + \mathbf{X}''\right]^{-1}_{\tilde{r}\tilde{r}_1}\tilde{\phi}'_{\tilde{r}_1}\tilde{\phi}'_{\tilde{r}_2}\left[\tilde{\mathbf{T}}' + \mathbf{X}''\right]^{-1}_{\tilde{r}_2\tilde{r}} + \left[\tilde{\mathbf{T}}' + \mathbf{X}''\right]^{-1}_{\tilde{r}\tilde{r}} - (\tilde{\mathbf{T}}')^{-1}_{\tilde{r}\tilde{r}} = 0\,, \tag{D.6}$$

where $\tilde{\mathbf{T}}' = \tilde{\mathbf{T}} + x_2 I$, $\mathbf{X}'' = \mathbf{X}' - x_2$ and $\tilde{\phi}'_{\tilde{r}} = \frac{m}{\sqrt{N}}\tilde{\phi}_{\tilde{r}}$. Writing $\mathbf{X}'' = \mathbf{X}^{(1)} + \mathbf{X}^{(2)} + \mathbf{X}^{(3)} + ..$, where $\mathbf{X}^{(k)}$ is the $k$-th order term in $\tilde{\phi}'_{\tilde{r}}\tilde{\phi}'_{\tilde{r}'}$, we obtain an infinite series of recursion relations that determines $\mathbf{X}^{(k)}$ in terms of $\mathbf{X}^{(n)}$ for $1 \le n \le k-1$. The first few equations read

$$\left[(\tilde{\mathbf{T}}')^{-1}\mathbf{X}^{(1)}(\tilde{\mathbf{T}}')^{-1}\right]_{\tilde{r}\tilde{r}} = \int d^D\tilde{r}_1 d^D\tilde{r}_2 (\tilde{\mathbf{T}}')^{-1}_{\tilde{r}\tilde{r}_1}(\tilde{\phi}'_{\tilde{r}_1}\tilde{\phi}'_{\tilde{r}_2})(\tilde{\mathbf{T}}')^{-1}_{\tilde{r}_2\tilde{r}}\,,$$

$$\left[(\tilde{\mathbf{T}}')^{-1}\mathbf{X}^{(2)}(\tilde{\mathbf{T}}')^{-1}\right]_{\tilde{r}\tilde{r}} = -2\int d^D\tilde{r}_1 d^D\tilde{r}_2\left[(\tilde{\mathbf{T}}')^{-1}\mathbf{X}^{(1)}(\tilde{\mathbf{T}}')^{-1}\right]_{\tilde{r}\tilde{r}_1}(\tilde{\phi}'_{\tilde{r}_1}\tilde{\phi}'_{\tilde{r}_2})\left[(\tilde{\mathbf{T}}')^{-1}\right]_{\tilde{r}_2\tilde{r}}$$
$$+\left[(\tilde{\mathbf{T}}')^{-1}\mathbf{X}^{(1)}(\tilde{\mathbf{T}}')^{-1}\mathbf{X}^{(1)}(\tilde{\mathbf{T}}')^{-1}\right]_{\tilde{r}\tilde{r}}\,,$$

$$\left[(\tilde{\mathbf{T}}')^{-1}\mathbf{X}^{(3)}(\tilde{\mathbf{T}}')^{-1}\right]_{\tilde{r}\tilde{r}} = -2\int d^D\tilde{r}_1 d^D\tilde{r}_2\left[(\tilde{\mathbf{T}}')^{-1}\mathbf{X}^{(2)}(\tilde{\mathbf{T}}')^{-1}\right]_{\tilde{r}\tilde{r}_1}(\tilde{\phi}'_{\tilde{r}_1}\tilde{\phi}'_{\tilde{r}_2})\left[(\tilde{\mathbf{T}}')^{-1}\right]_{\tilde{r}_2\tilde{r}}$$
$$+\int d^D\tilde{r}_1 d^D\tilde{r}_2\left[(\tilde{\mathbf{T}}')^{-1}\mathbf{X}^{(1)}(\tilde{\mathbf{T}}')^{-1}\right]_{\tilde{r}\tilde{r}_1}(\tilde{\phi}'_{\tilde{r}_1}\tilde{\phi}'_{\tilde{r}_2})\left[(\tilde{\mathbf{T}}')^{-1}\mathbf{X}^{(1)}(\tilde{\mathbf{T}}')^{-1}\right]_{\tilde{r}_2\tilde{r}}$$
$$+2\int d^D\tilde{r}_1 d^D\tilde{r}_2\left[(\tilde{\mathbf{T}}')^{-1}\mathbf{X}^{(1)}(\tilde{\mathbf{T}}')^{-1}\mathbf{X}^{(1)}(\tilde{\mathbf{T}}')^{-1}\right]_{\tilde{r}\tilde{r}_1}(\tilde{\phi}'_{\tilde{r}_1}\tilde{\phi}'_{\tilde{r}_2})\left[(\tilde{\mathbf{T}}')^{-1}\right]_{\tilde{r}_2\tilde{r}}$$
$$+2\left[(\tilde{\mathbf{T}}')^{-1}\mathbf{X}^{(2)}(\tilde{\mathbf{T}}')^{-1}\mathbf{X}^{(1)}(\tilde{\mathbf{T}}')^{-1}\right]_{\tilde{r}\tilde{r}}$$
$$-\left[(\tilde{\mathbf{T}}')^{-1}\mathbf{X}^{(1)}(\tilde{\mathbf{T}}')^{-1}\mathbf{X}^{(1)}(\tilde{\mathbf{T}}')^{-1}\mathbf{X}^{(1)}(\tilde{\mathbf{T}}')^{-1}\right]_{\tilde{r}\tilde{r}}\,. \tag{D.7}$$

Let us solve $\mathbf{X}^{(1)}$ explicitly. If we view $X_{\tilde{r}}^{(1)}$ as the $\tilde{r}$-th element of a vector $\vec{X}^{(1)}$, the first equation in Eq. (D.7) is written as $\mathbf{L}\vec{X}^{(1)} = \vec{\Phi}$, where

$$
\mathbf{L}_{\tilde{r}\tilde{r}'} = \left[(\tilde{\mathbf{T}}')_{\tilde{r}\tilde{r}'}^{-1}\right]^2 = \left[\int \frac{d^D\tilde{Q}}{(2\pi)^D} \frac{e^{i\tilde{Q}(\tilde{r}-\tilde{r}')}}{\tilde{Q}^2/m^2 + \tilde{\Sigma}}\right]^2 ,
$$

$$
\vec{\Phi}_{\tilde{r}} = \left[\int d^D\tilde{r}'(\tilde{\mathbf{T}}')_{\tilde{r}\tilde{r}'}^{-1}\tilde{\phi}'_{\tilde{r}'}\right]^2 = \left[\int d^D\tilde{r}' \int \frac{d^D\tilde{Q}}{(2\pi)^D} \frac{e^{i\tilde{Q}(\tilde{r}-\tilde{r}')}}{\tilde{Q}^2/m^2 + \tilde{\Sigma}}\tilde{\phi}'_{\tilde{r}'}\right]^2 . \tag{D.8}
$$

The solution can be obtained by inverting $\mathbf{L}$ as

$$
X_{\tilde{r}}^{(1)} = \frac{m^2}{N} \frac{1}{\frac{\Gamma(2-\frac{D}{2})}{(4\pi)^{D/2}}\int_0^1 du\left[-u(1-u)\tilde{\nabla}^2/m^2 + \tilde{\Sigma}\right]^{\frac{D}{2}-2}}\left[\frac{1}{-\tilde{\nabla}^2/m^2 + \tilde{\Sigma}}\tilde{\phi}_{\tilde{r}}\right]^2 , \tag{D.9}
$$

where the answer is written in terms of $\tilde{\phi}_{\tilde{r}}$. It is noted that $\frac{1}{N}\mathbf{L}_{\tilde{r}\tilde{r}'}^{-1}$ actually gives the propagator of singlet field in the large N limit while $\tilde{\mathbf{T}}_{\tilde{r}\tilde{r}'}^{-1}$ is the free propagator. One can use Feynman diagrams to represent $X_{\tilde{r}}^{(k)}$ as is shown in Fig. 1.

# E  Leading scaling operators at the Wilson-Fisher fixed point

The Wilson-Fisher fixed point has two relevant scaling operators. In this section, we compute them in $D < 4$. In order to extract a scaling operator with a certain symmetry charge, we perturb the UV action with an operator that has the same symmetry charge, and study the RG flow induced by it. The RG flow can be decomposed into eigen-modes each of which has a definite scaling dimension.

## E.1  O(N) singlet

To extract the leading singlet operator, let us perturb the system with a uniform mass : $-\frac{m^2}{2}\mathbf{M}'_{ij} = -\frac{m^2}{2}\mathbf{M}^c_{ij} + \epsilon'\delta_{ij}$, where $-\frac{m^2}{2}\mathbf{M}^c$ is a hopping matrix that flows to the Wilson-Fisher fixed point. In the presence of the perturbation, the saddle point solution of $\mathbf{X}$ is modified to $\mathbf{X} + \delta\mathbf{X}$. However, the variation of $\mathbf{X}$ does not contribute to the change of the effective action to the linear order in $\epsilon'$ because the effective action is stationary with respect to $\mathbf{X}$ at the saddle point. As a result, the variation of the effective action caused by the change in $\mathbf{M}$ becomes

$$
\begin{aligned}
\Delta_\epsilon S_{tot} &= -\frac{2}{m^2}\epsilon'\sum_i \partial_{M_{ii}}S_{tot} \\
&= -\frac{2}{m^2}\epsilon'\int d^D r\left(\frac{e^{2z^*}}{(e^{2z^*}-1)^2} \times \int d^D r_1 d^D r_2\, m^2\, [\mathbf{K}+\mathbf{X}]_{rr_1}^{-1}(\phi_{r_1}\phi_{r_2})[\mathbf{K}+\mathbf{X}]_{r_2r}^{-1} + N[\mathbf{K}+\mathbf{X}]_{rr}^{-1}\right) \\
&= -\epsilon'\frac{m^2 N}{2\lambda}\int d^D\tilde{r}\left[e^{Dz^*}x_0 + e^{(D-2)z^*}X'_{\tilde{r}}\right]. \tag{E.1}
\end{aligned}
$$

From the second line to the last line, the saddle point equation in Eq. (10) is used. The final answer is expressed in terms of the rescaled coordinate, $\tilde{r} = re^{-z^*}$. The first term is the correction to the identity operator with scaling dimension 0. It describes the change of the free energy. The second term shows that $X'_{\tilde{r}}$ is the next leading singlet scaling operator with the scaling dimension $\Delta_X = 2$.

(1)

(2)

(3)

Figure 1: The Feynman diagrams representing the solution $X_{\tilde{r}}^{(k)}$ in Eq. (D.7). Black dots denote $\tilde{\phi}_{\tilde{r}}$. The solid lines represent the free propagators of $\phi$, $\tilde{\mathbf{T}}_{\tilde{r}\tilde{r}'}^{-1}$. The wave lines are the propagators of the singlet field, $-\frac{1}{N}\mathbf{L}_{\tilde{r}\tilde{r}'}^{-1}$. There is an extra minus sign for each propagator in red.

## E.2 $O(N)$ non-singlet

In order to extract the scaling operator in the fundamental representation of $O(N)$, we add a non-local hopping term between the origin and $R$ in the $R \to \infty$ limit. This is equivalent to inserting two fundamental fields far from each other. For this, we consider a deformation given by $-\frac{m^2}{2}\mathbf{M}'_{rr'} = -\frac{m^2}{2}\mathbf{M}^c_{rr'} + \epsilon\left[\delta(r)\delta(r'-R) + \delta(r')\delta(r-R)\right]$. In the large $\tilde{R}$ limit, the change in the IR action becomes

$$\Delta_\epsilon S_{tot} = -\frac{4}{m^2}\epsilon\left(\frac{e^{2z^*}}{(e^{2z^*}-1)^2}\frac{m^2}{2}\int d^D r_1 d^D r_2\,[\mathbf{K}+\mathbf{X}]^{-1}_{0r_1}(\phi_{r_1}\phi_{r_2})[\mathbf{K}+\mathbf{X}]^{-1}_{r_2 R} + \frac{N}{2}[\mathbf{K}+\mathbf{X}]^{-1}_{0R}\right)$$

$$= -2\epsilon e^{(2-D)z^*}\int d^D \tilde{r}_1 d^D \tilde{r}_2\,\left[\tilde{\mathbf{T}}+\mathbf{X}'\right]^{-1}_{0\tilde{r}_1}(\tilde{\phi}_{\tilde{r}_1}\tilde{\phi}_{\tilde{r}_2})\left[\tilde{\mathbf{T}}+\mathbf{X}'\right]^{-1}_{\tilde{r}_2 \tilde{R}}. \tag{E.2}$$

In the last line, $\lim_{\tilde{R}\to\infty}\left[\tilde{\mathbf{T}}+\mathbf{X}'\right]^{-1}_{0\tilde{R}} = 0$ is used for which large $R$ limit is taken first before the large $z$ limit. In this limit, $\Delta_\epsilon S_{tot}$ can be viewed as two local operators each of which inserted at the origin and the infinity, respectively. This shows that $\phi^S_{\tilde{r}} = \int d^D \tilde{r}_1 [\tilde{\mathbf{T}}+\mathbf{X}']^{-1}_{\tilde{r}\tilde{r}_1}\tilde{\phi}_{\tilde{r}_1}$ is the scaling operator in the fundamental representation of $O(N)$ with scaling dimension $\Delta_\phi = \frac{D-2}{2}$.

# F  Rescaling of spacetime and field

In this appendix, we elaborate on the derivations of Eqs. (12) and (13) from Eqs. (9) and (10). For convenience, the lattice spacing is chosen to be 1 at UV. Matrix multiplications in real space are written as integrations as $(\mathbf{AB})_{r_1 r_2} = \sum_r A_{r_1 r}B_{rr_2} = \int dr A_{r_1 r}B_{rr_2}$ in which the identity matrix becomes $I_{rr'} = \delta(r-r')$. In order to make it manifest that the effective action is scale invariant at the critical point, we use the rescaled momentum, coordinate and field defined by $\tilde{Q} = Qe^{z^*}$, $\tilde{r} = re^{-z^*}$ and $\tilde{\phi}_{\tilde{r}} = \phi_r e^{\frac{D}{2}z^*}$. Taking the large $z$ limit with fixed $\tilde{Q}$ is tantamount to zooming in toward the neighbourhood of $Q = 0$ which contains the dynamical information on the universal long-distance physics.

The kernel in the first term of Eq. (9) is written as

$$\left[(e^{2z^*}-1)(\mathbf{K}+\mathbf{X})\right]^{-1}_{rr'} = \left[(e^{2z^*}-1)(\mathbf{K}+x_0 I) + (e^{2z^*}-1)(\mathbf{X}-x_0 I)\right]^{-1}_{rr'}. \tag{F.1}$$

Treating

$$\tilde{\mathbf{T}}^{-1}_{\tilde{r}\tilde{r}'} \equiv e^{Dz^*}\left[(e^{2z^*}-1)(\mathbf{K}+x_0 I)\right]^{-1}_{rr'} = \int^{\Lambda e^{z^*}}\frac{d^D \tilde{Q}}{(2\pi)^D}\frac{e^{i\tilde{Q}(\tilde{r}-\tilde{r}')}}{\tilde{Q}^2/m^2 + 1} \tag{F.2}$$

as the zero-th order term, one can expand Eq. (F.1) in powers of $\mathbf{X}'_{\tilde{r}\tilde{r}'} \equiv e^{2z^*}\delta(\tilde{r}-\tilde{r}')(X_r - x_0)$ as

$$e^{-Dz^*}\left\{\tilde{\mathbf{T}}^{-1}_{\tilde{r}\tilde{r}'} - \sum_{M_1 \geq 1}^{\infty}(-1)^{M_1-1}\int d^D \tilde{R}\,\tilde{\mathbf{T}}^{-1}_{\tilde{r}\tilde{R}}\mathbf{X}'_{\tilde{R}\tilde{R}}\int d^D \tilde{R}_1\,\tilde{\mathbf{T}}^{-1}_{\tilde{R}\tilde{R}_1}\mathbf{X}'_{\tilde{R}_1 \tilde{R}_1}\int d^D \tilde{R}_2\,\tilde{\mathbf{T}}^{-1}_{\tilde{R}_1 \tilde{R}_2}\right.$$

$$\left.\times \cdots \times \int d^D \tilde{R}_{M_1-1}\tilde{\mathbf{T}}^{-1}_{\tilde{R}_{M_1-2}\tilde{R}_{M_1-1}}\mathbf{X}'_{\tilde{R}_{M_1-1}\tilde{R}_{M_1-1}}\tilde{\mathbf{T}}^{-1}_{\tilde{R}_{M_1-1}\tilde{r}'}\right\} = e^{-Dz^*}\left[\tilde{\mathbf{T}}+\mathbf{X}'\right]^{-1}_{\tilde{r}\tilde{r}'}. \tag{F.3}$$

Consequently,

$$\left[(e^{2z^*}-1)(\mathbf{K}+\mathbf{X})\right]^{-1}_{rr'} = e^{-Dz^*}\left[\tilde{\mathbf{T}}+\mathbf{X}'\right]^{-1}_{\tilde{r}\tilde{r}'}. \tag{F.4}$$

The second term of Eq. (9) can be written as

$$\sum_r \log\left[(e^{2z^*}-1)(\mathbf{K}+\mathbf{X})\right]_{rr} = \sum_r \log[I+A], \tag{F.5}$$

where $A = (e^{2z^*} - 1)(\mathbf{K} + \mathbf{X}) - I$. An expansion in powers of $A$ gives

$$\sum_r \log[I + A] = \sum_{M' \geq 1} \frac{(-1)^{M'-1}}{M'} \sum_{r_1,..,r_{M'}} A_{r_1 r_2}..A_{r_{M'} r_1}. \tag{F.6}$$

In the rescaled coordinate, the elements of $A$ becomes $A_{rr'} = e^{-Dz^*}\left[(\tilde{\mathbf{T}} + \mathbf{X}')_{\tilde{r}\tilde{r}'} - \delta(\tilde{r} - \tilde{r}')\right]$ and $\sum_r = e^{Dz^*} \int d^D \tilde{r}$. From this, we obtain

$$\sum_r \left\{\log\left[(e^{2z^*} - 1)(\mathbf{K} + \mathbf{X})\right]\right\}_{rr} = \int d^D \tilde{r} \left\{\log\left[\tilde{\mathbf{T}} + \mathbf{X}'\right]\right\}_{\tilde{r}\tilde{r}}. \tag{F.7}$$

Using Eq. (F.2), Eq. (F.4) and Eq. (F.7), we can obtain Eq. (12) and Eq. (13) from Eq. (9) and Eq. (10).

# G  Correlation functions from generating function

The $n$-point function can be expressed as derivatives of the generating function,

$$\langle \phi_{r_1}^{a_1} \phi_{r_2}^{a_2} \cdots \phi_{r_n}^{a_n} \rangle = -\frac{\delta^n W[J]}{\delta J_{r_1}^{a_1} \delta J_{r_2}^{a_2} \cdots \delta J_{r_n}^{a_n}}\bigg|_{J=0}, \tag{G.1}$$

where the full generating function is given by the effective action in the large $z$ limit as is shown in Eq. (24). In this appendix, we explicitly compute the 2-point function and the 4-point function using the exact effective action in Eq. (9).

## G.1  2-pt correlation function

The generating function is a function of $J_i^a$ and $X_k[J]$, where $X_k[J]$ satisfies the saddle point equation in Eq. (10). The derivative of $W$ with respect to a source is written as $\frac{\delta W[J]}{\delta J_i^a} = \frac{\partial W}{\partial J_i^a} + \sum_k \frac{\partial W}{\partial X_k} \frac{\partial X_k}{\partial J_i^a}\big|_{X=\bar{X}}$, and the 2-pt correlation function can be expressed as

$$
\begin{aligned}
G_2^{ab}[r_1, r_2] &= -\frac{\delta^2 W}{\delta J_{r_1}^a \delta J_{r_2}^b}\bigg|_{J=0} = -\left[\frac{\partial}{\partial J_{r_1}^a} + \sum_r \frac{\partial X_r}{\partial J_{r_1}^a} \frac{\partial}{\partial X_r}\right]\left[\frac{\partial W}{\partial J_{r_2}^b} + \sum_{r'} \frac{\partial W}{\partial X_{r'}} \frac{\partial X_{r'}}{\partial J_{r_2}^b}\right]_{\substack{J=0 \\ X=\bar{X}}} \\
&= -\left[\frac{\partial^2 W}{\partial J_{r_1}^a \partial J_{r_2}^b} + \sum_r \frac{\partial^2 W}{\partial J_{r_1}^a \partial X_r} \frac{\partial X_r}{\partial J_{r_2}^b} + \sum_r \frac{\partial X_r}{\partial J_{r_1}^a} \frac{\partial^2 W}{\partial X_r \partial J_{r_2}^b} + \sum_{rr'} \frac{\partial^2 W}{\partial X_r \partial X_{r'}} \frac{\partial X_r}{\partial J_{r_1}^a} \frac{\partial X_{r'}}{\partial J_{r_2}^b}\right. \\
&\qquad + \left.\sum_r \frac{\partial W}{\partial X_r} \frac{\partial^2 X_r}{\partial J_{r_1}^a \partial J_{r_2}^b}\right]_{\substack{J=0 \\ X=\bar{X}}} \\
&= -\left[\frac{\partial^2 W}{\partial J_{r_1}^a \partial J_{r_2}^b} + \sum_r \frac{\partial W}{\partial X_r} \frac{\partial^2 X_r}{\partial J_{r_1}^a \partial J_{r_2}^b}\right]_{\substack{J=0 \\ X=\bar{X}}}. \tag{G.2}
\end{aligned}
$$

In the last equality, we used the fact that both $W$ and $X$ are even functions of $J_r^a$. Besides, we also have $\frac{\partial W}{\partial X_l}\big|_{X=\bar{X}} = 0$ because $\bar{X}_l$ satisfies the saddle point equation of $W$. Thus, only the first term contributes to the 2-pt correlation of $\phi$,

$$\frac{\partial^2 W}{\partial J_{r_1}^a \partial J_{r_2}^b} = -\frac{\delta_{ab}}{m_z^2} \delta_{r_1, r_2} + \frac{\partial^2 S_1^z[e^z J/m_z^2]}{\partial J_{r_1}^a \partial J_{r_2}^b} = -\frac{\delta_{ab}}{m^2}[\mathbf{K} + \mathbf{X}]_{r_1, r_2}^{-1}. \tag{G.3}$$

Setting $J_r^a = 0$ and $\mathbf{X} = x(z)I$ at the critical point with $x(z) + a + 1 = x_2 e^{-2z}$, we obtain

$$G_2^{ab}[r_1, r_2] = \frac{\delta_{ab}}{m^2} \int \frac{d^D Q}{(2\pi)^D} \frac{e^{iQ \cdot (r_1 - r_2)}}{\frac{Q^2}{m^2} + x_2 e^{-2z}} \xrightarrow{z \to \infty} \frac{\delta_{ab}}{|r_1 - r_2|^{D-2}} \, . \tag{G.4}$$

This is the free propagator of the fundamental field as expected in the large $N$ limit.

## G.2  4-pt correlation function

The 3-point function is zero since it does not respect the O(N) symmetry. The 4-point function is expressed as

$$G_4^{abcd}[r_1, r_2, r_3, r_4] = - \left. \frac{\delta^4 W}{\delta J_{r_1}^a \delta J_{r_2}^b \delta J_{r_3}^c \delta J_{r_4}^d} \right|_{J=0}, \tag{G.5}$$

where

$$
\begin{aligned}
\frac{\delta^4 W}{\delta J_{r_1}^a \delta J_{r_2}^b \delta J_{r_3}^c \delta J_{r_4}^d} =& \left[ \frac{\partial}{\partial J_{r_4}^d} + \sum_r \frac{\partial X_r}{\partial J_{r_4}^d} \frac{\partial}{\partial X_r} \right] \left[ \frac{\partial}{\partial J_{r_3}^c} + \sum_r \frac{\partial X_r}{\partial J_{r_3}^c} \frac{\partial}{\partial X_r} \right] \left[ \frac{\partial}{\partial J_{r_2}^b} + \sum_r \frac{\partial X_r}{\partial J_{r_2}^b} \frac{\partial}{\partial X_r} \right] \\
&\times \left[ \frac{\partial W}{\partial J_{r_1}^a} + \sum_r \frac{\partial W}{\partial X_r} \frac{\partial X_r}{\partial J_{r_1}^a} \right]_{X=\bar{X}} \\
=& \sum_r \left[ \frac{\partial^3 W}{\partial X_r \partial J_{r_3}^c \partial J_{r_4}^d} \frac{\partial^2 X_r}{\partial J_{r_1}^a \partial J_{r_2}^b} + \frac{\partial^3 W}{\partial X_r \partial J_{r_2}^b \partial J_{r_4}^d} \frac{\partial^2 X_r}{\partial J_{r_1}^a \partial J_{r_3}^c} \right. \\
&+ \frac{\partial^3 W}{\partial X_r \partial J_{r_2}^b \partial J_{r_3}^c} \frac{\partial^2 X_r}{\partial J_{r_1}^a \partial J_{r_4}^d} + \frac{\partial^3 W}{\partial X_r \partial J_{r_1}^a \partial J_{r_2}^b} \frac{\partial^2 X_r}{\partial J_{r_3}^c \partial J_{r_4}^d} \\
&+ \frac{\partial^3 W}{\partial X_r \partial J_{r_1}^a \partial J_{r_3}^c} \frac{\partial^2 X_r}{\partial J_{r_2}^b \partial J_{r_4}^d} + \frac{\partial^3 W}{\partial X_r \partial J_{r_1}^a \partial J_{r_4}^d} \frac{\partial^2 X_r}{\partial J_{r_2}^b \partial J_{r_3}^c} \\
&+ \sum_{r'} \left( \frac{\partial^2 W}{\partial X_r \partial X_{r'}} \frac{\partial^2 X_r}{\partial J_{r_1}^a \partial J_{r_3}^c} \frac{\partial^2 X_{r'}}{\partial J_{r_2}^b \partial J_{r_4}^d} + \frac{\partial^2 W}{\partial X_r \partial X_{r'}} \frac{\partial^2 X_r}{\partial J_{r_1}^a \partial J_{r_4}^d} \frac{\partial^2 X_{r'}}{\partial J_{r_2}^b \partial J_{r_3}^c} \right. \\
&\left. \left. + \frac{\partial^2 W}{\partial X_r \partial X_{r'}} \frac{\partial^2 X_r}{\partial J_{r_1}^a \partial J_{r_2}^b} \frac{\partial^2 X_{r'}}{\partial J_{r_3}^c \partial J_{r_4}^d} \right) \right]_{X=\bar{X}}, \tag{G.6}
\end{aligned}
$$

where the terms that vanish at $J_r^a = 0$ are dropped. Defining $\mathbb{L}$ to be a matrix whose element is given by $\mathbb{L}_{rr'} = [(\mathbf{K} + xI)_{rr'}^{-1}]^2 + \frac{m^4}{4\lambda} \delta_{rr'}$, we write

$$\left. \frac{\partial^2 X_r}{\partial J_{r_1}^a \partial J_{r_2}^b} \right|_{\substack{J=0 \\ X=\bar{X}}} = \frac{2\delta_{ab}}{Nm^2} \sum_{r'} \mathbb{L}_{rr'}^{-1} (\mathbf{K} + xI)_{r'r_1}^{-1} (\mathbf{K} + xI)_{r_2 r'}^{-1} \tag{G.7}$$

$$\left. \frac{\partial^3 W}{\partial X_r \partial J_{r_1}^a \partial J_{r_2}^b} \right|_{\substack{J=0 \\ X=\bar{X}}} = \frac{\delta_{ab}}{m^2} (\mathbf{K} + xI)_{r_1, r}^{-1} (\mathbf{K} + xI)_{r, r_2}^{-1}, \tag{G.8}$$

$$\left. \frac{\partial^2 W}{\partial X_r \partial X_{r'}} \right|_{\substack{J=0 \\ X=\bar{X}}} = -\frac{N}{2} \mathbb{L}_{rr'}, \tag{G.9}$$

from Eq. (9) and Eq. (10). In the momentum space, $\mathbb{L}$ can be written as $\mathbb{L}_{rr'} = \int \frac{d^D q}{(2\pi)^D} e^{iq \cdot (r-r')} \mathbb{L}_q$, where $\mathbb{L}_q = c_1 \int_0^1 du \, [u(1-u)\frac{q^2}{m^2} + x_2 e^{-2z}]^{\frac{D}{2}-2} + \frac{m^4}{4\lambda} = c_2 \frac{q^{D-4}}{m^{D-4}} + \frac{m^4}{4\lambda}$ in the large $z$ limit with $c_1 = (4\pi)^{-D/2} \Gamma[2 - \frac{D}{2}]$ and $c_2 = \frac{\Gamma[\frac{4-D}{2}] \Gamma[\frac{D-2}{2}]}{2^{2D-3} \pi^{\frac{D-1}{2}} \Gamma[\frac{D-1}{2}]}$. Then, two different terms that appear Eq. (G.6) are given by

$$-\sum_r \frac{\partial^3 W}{\partial X_r \partial J_{r_3}^c \partial J_{r_4}^d} \frac{\partial^2 X_r}{\partial J_{r_1}^a \partial J_{r_2}^b}\Big|_{\substack{J=0 \\ X=\bar{X}}} = -\frac{2\delta_{ab}\delta_{cd}}{Nm^4}$$

$$\times \sum_{rr'} \mathbb{L}_{rr'}^{-1}(\mathbf{K}+xI)_{r_3,r}^{-1}(\mathbf{K}+xI)_{r,r_4}^{-1}(\mathbf{K}+xI)_{r'r_1}^{-1}(\mathbf{K}+xI)_{r_2r'}^{-1}$$

$$= \frac{2\delta_{ab}\delta_{cd}}{Nm^4} I^z[r_1,r_2,r_3,r_4],$$

$$\sum_{r,r'} \frac{\partial^2 W}{\partial X_r \partial X_{r'}} \frac{\partial^2 X_r}{\partial J_{r_1}^a \partial J_{r_2}^c} \frac{\partial^2 X_{r'}}{\partial J_{r_3}^b \partial J_{r_4}^d} = \frac{2\delta_{ab}\delta_{cd}}{Nm^4} I^z[r_1,r_2,r_3,r_4], \tag{G.10}$$

where

$$I^z[r_1,r_2,r_3,r_4] = $$


$$= -\int d^D r\, d^D r' \frac{1}{|r_1-r'|^{D-2}} \frac{1}{|r_2-r'|^{D-2}} \frac{1}{|r-r'|^4} \frac{1}{|r_3-r|^{D-2}} \frac{1}{|r_4-r|^{D-2}}. \tag{G.11}$$

Note that $I^z[r_1,r_2,r_3,r_4]$ is a product of four free propagator and a propagator of the O(N) singlet, and invariant under the interchange of the first two and the last two coordinates, $I^z[r_1,r_2,r_3,r_4] = I^z[r_3,r_4,r_1,r_2]$. In the momentum space, it can be expressed as

$$I^z[r_1,r_2,r_3,r_4]$$

$$= -\int \frac{d^D q}{(2\pi)^D} \frac{d^D p}{(2\pi)^D} \frac{d^D Q}{(2\pi)^D} \mathbb{L}_Q^{-1} \frac{e^{iq\cdot(r_3-r_4)}}{\frac{q^2}{m^2}+x_2 e^{-2z}} \frac{e^{iQ\cdot(r_4-r_2)}}{\frac{(q-Q)^2}{m^2}+x_2 e^{-2z}} \frac{e^{ip\cdot(r_2-r_1)}}{\frac{p^2}{m^2}+x_2 e^{-2z}} \frac{1}{\frac{(p-Q)^2}{m^2}+x_2 e^{-2z}}$$

$$= \frac{m^{8-D}}{4(2\pi)^D} \int \frac{d^D Q}{(2\pi)^D} \frac{1}{c_2 \frac{Q^{D-4}}{m^{D-4}}+\frac{m^4}{4\lambda}} \int_0^1 dx \int_0^1 dy \left(\frac{\sqrt{x(1-x)}\sqrt{y(1-y)}|Q|^2}{r_{12}r_{34}}\right)^{\frac{D-4}{2}}$$

$$\times K_{\frac{D-4}{2}}\left(\sqrt{x(1-x)}|Q|r_{34}\right) K_{\frac{D-4}{2}}\left(\sqrt{y(1-y)}|Q|r_{12}\right) e^{i|Q|r_{13}\cos\theta_{13}} e^{ix|Q|r_{34}\cos\theta_{34}} e^{-iy|Q|r_{12}\cos\theta_{12}}, \tag{G.12}$$

where $K_\nu(z)$ is the modified Bessel functions of the second kind. In the above expression, $r_i - r_j = r_{ij}\hat{r}_{ij}$ where $\hat{r}_{ij}$ is a unit vector connecting points $i$ and $j$. $Q\cdot\hat{r}_{ij} = |Q||r_{ij}|\cos\theta_{ij}$, where $\theta_{ij}$ is the angle between $Q$ and $\hat{r}_{ij}$. Together with other terms from permutation, we can express the connected 4-pt correlation function as

$$G_4^{abcd}[r_1,r_2,r_3,r_4]$$

$$= \frac{2}{Nm^4}\left[\delta_{ab}\delta_{cd} I^z[r_1,r_2,r_3,r_4] + \delta_{ac}\delta_{bd} I^z[r_1,r_3,r_2,r_4] + \delta_{ad}\delta_{bc} I^z[r_1,r_4,r_2,r_3]\right]. \tag{G.13}$$

In general, the connected 4-point function can be written as

$$G_4^{abcd}[r_1,r_2,r_3,r_4] = f^{abcd}\left(\frac{r_{12}r_{34}}{r_{13}r_{24}}, \frac{r_{12}r_{34}}{r_{14}r_{23}}\right) \prod_{i>j} r_{ij}^{-\frac{D-2}{3}}, \tag{G.14}$$

where $r_{ij} = |r_i - r_j|$ and $f^{abcd}(u,v)$ is a universal function of the cross ratios. As a concrete example, let us consider a case where the four points lie on a line with $r_{12} = r_{34} = \alpha r$ and

$r_{14} = r_{23} = r$. $\alpha$ is a dimensionless parameter that determines the cross ratios. In this case, the 4-point function in the large $r$ limit becomes

$$G_4^{abcd}[r_1, r_2, r_3, r_4] = \frac{2}{N} \frac{1}{r^{2D-4}} (F_1[\alpha] + F_2[\alpha] + F_3[\alpha]) , \qquad (G.15)$$

where

$$F_1[\alpha] = -\frac{1}{4(2\pi)^D c_2} \frac{1}{\alpha^{D-4}} \int \frac{d^D Q'}{(2\pi)^D} \int_0^1 dx \int_0^1 dy \left(\sqrt{x(1-x)}\sqrt{y(1-y)}\right)^{\frac{D-4}{2}}$$
$$\times K_{\frac{D-4}{2}}\left(\sqrt{x(1-x)}\alpha|Q'|\right) K_{\frac{D-4}{2}}\left(\sqrt{y(1-y)}\alpha|Q'|\right) e^{i[(x-y)\alpha+1]|Q'|\cos\theta} ,$$

$$F_2[\alpha] = -\frac{1}{4(2\pi)^D c_2} \frac{1}{(1-\alpha^2)^{\frac{D-4}{2}}} \int \frac{d^D Q'}{(2\pi)^D} \int_0^1 dx \int_0^1 dy \left(\sqrt{x(1-x)}\sqrt{y(1-y)}\right)^{\frac{D-4}{2}}$$
$$\times K_{\frac{D-4}{2}}\left(\sqrt{x(1-x)}(1+\alpha)|Q'|\right) K_{\frac{D-4}{2}}\left(\sqrt{y(1-y)}(1-\alpha)|Q'|\right)$$
$$\times e^{i[\alpha+x(1+\alpha)+y(1-\alpha)]|Q'|\cos\theta} ,$$

$$F_3[\alpha] = -\frac{1}{4(2\pi)^D c_2} \int \frac{d^D Q'}{(2\pi)^D} \int_0^1 dx \int_0^1 dy \left(\sqrt{x(1-x)}\sqrt{y(1-y)}\right)^{\frac{D-4}{2}}$$
$$\times K_{\frac{D-4}{2}}\left(\sqrt{x(1-x)}|Q'|\right) K_{\frac{D-4}{2}}\left(\sqrt{y(1-y)}|Q'|\right) e^{i(\alpha+x+y)|Q'|\cos\theta} , \qquad (G.16)$$

for $D < 4$. Similarly, we can compute other $n$-point functions from the exact effective action.

# H Effective action from an alternative RG scheme

In this section, we derive the fixed point effective action in an alternative RG scheme, where the free massless theory is used as the reference theory. The partition function is still written as the overlap between two states as $Z = \langle S'_{ref} | S'_1 \rangle$. The state associated with the new reference theory reads

$$|S'_{ref}\rangle = \int \mathcal{D}\phi \; e^{-\frac{1}{2} \int d^D k \; G_M^{-1}(k)\phi_k\phi_{-k}} |\phi\rangle , \qquad (H.1)$$

where $k$ is momentum and $G_M^{-1}(k) = e^{\frac{k^2}{M^2}} k^2$ is a regularized kinetic term that suppresses modes with momenta larger than UV cutoff $M$. The deformation includes the bi-linear operators and the on-site quartic interaction, $S'_1 = \sum_{ij} J_{ij}\phi_i\phi_j + \frac{\lambda}{N} \sum_i (\phi_i^2)^2$. The state associated with the deformation at $z = 0$ is

$$|S_1'^0\rangle = \int \mathcal{D}t^0 \mathcal{D}p^0 \; e^{-N S_{UV}[t^0, p^0]} |t^0\rangle , \qquad (H.2)$$

where the UV boundary action is $S_{UV} = \int d^D r d^D r' \, (it_{rr'}^0 + J_{rr'}) p_{rr'}^0 + \lambda \int d^D r \, (p_{rr}^0)^2$, and $|t\rangle$ is defined in Eq. (5). The RG Hamiltonian that satisfies $\hat{H}^\dagger |S_{ref}\rangle = 0$ is given by [2, 29]

$$\hat{H} = \int d^D k \left[ \frac{\tilde{G}(k)}{2} \hat{\pi}_k \hat{\pi}_{-k} - i \left( \frac{D+2}{2} \hat{\phi}_k + k\partial_k \hat{\phi}_k \right) \hat{\pi}_k + C \right] , \qquad (H.3)$$

where $\tilde{G}(k) = \frac{\partial G_M(k)}{\partial \ln M} = \frac{2}{M^2} e^{-\frac{k^2}{M^2}}$ and $C = -\int d^D k \delta^D(0) \left[ \frac{\tilde{G}}{2} G_M^{-1} + 1 \right]$ is a constant. $\hat{\pi}$ is the canonical conjugate of $\hat{\phi}$ satisfying $[\hat{\phi}_k, \hat{\pi}_{k'}] = i(2\pi)^D \delta^D(k-k')$. The first term in the Hamiltonian effectively integrates out modes with momenta between $Me^{-dz}$ and $M$. The second

term rescales the momentum and field as $k = \tilde{k}e^{-dz}$ and $\phi_k = e^{\frac{D+2}{2}dz}\phi_{\tilde{k}}$. Since the RG Hamiltonian is diagonal in the momentum space, it is convenient to express the basis states in Eq. (5) in the momentum space as

$$
|t\rangle = \int \mathcal{D}\phi \, e^{i \int d^D k_1 d^D k_2 t_{k_1,k_2} \phi_{k_1} \phi_{k_2}} |\phi\rangle, \tag{H.4}
$$

where $t_{r_1,r_2} = \int d^D k_1 d^D k_2 t_{k_1,k_2} e^{ik_1 \cdot r_1 + ik_2 \cdot r_2}$ and $\phi_r = \int \frac{d^D k}{(2\pi)^D} \phi_k e^{-ik \cdot r}$.

An infinitesimal RG transformation applied to the basis state at $z$ can be written as a linear superposition of the basis states at $z + dz$ as

$$
e^{-\hat{H}dz}|t^z\rangle = \int \mathcal{D}t^{z+dz}_{\tilde{k}_1,\tilde{k}_2} \mathcal{D}p^{z+dz}_{\tilde{k}_1,\tilde{k}_2} e^{-N\mathcal{L}^{z+dz}_{bulk}[t^{z+dz}_{\tilde{k}_1,\tilde{k}_2}, p^{z+dz}_{\tilde{k}_1,\tilde{k}_2}]dz} |t^{z+dz}\rangle, \tag{H.5}
$$

where

$$
\begin{aligned}
\mathcal{L}_{bulk} &= i \int d^D \tilde{k}_1 d^D \tilde{k}_2 \, (\partial_z t_{\tilde{k}_1,\tilde{k}_2}) p_{\tilde{k}_1,\tilde{k}_2} - i \int d^D \tilde{k} \, \tilde{G}(\tilde{k}) t_{\tilde{k},-\tilde{k}} \\
&+ 2 \int d^D \tilde{k} d^D \tilde{k}_1 d^D \tilde{k}_2 \, \tilde{G}(\tilde{k}) t_{\tilde{k}_1,-\tilde{k}} t_{\tilde{k}_2,\tilde{k}} p_{\tilde{k}_1,\tilde{k}_2} - i \int d^D \tilde{k}_1 d^D \tilde{k}_2 \, (2-D) t_{\tilde{k}_1,\tilde{k}_2} p_{\tilde{k}_1,\tilde{k}_2} \\
&+ i \int d^D \tilde{k}_1 d^D \tilde{k}_2 \, \tilde{k}_1 \left( \partial_{\tilde{k}_1} t_{\tilde{k}_1,\tilde{k}_2} \right) p_{\tilde{k}_1,\tilde{k}_2} + i \int d^D \tilde{k}_1 d^D \tilde{k}_2 \, \tilde{k}_2 \left( \partial_{\tilde{k}_2} t_{\tilde{k}_1,\tilde{k}_2} \right) p_{\tilde{k}_1,\tilde{k}_2}. \tag{H.6}
\end{aligned}
$$

Here we use $\tilde{k}_i$ for momentum to denote the fact that momentum has been already scaled through the RG Hamiltonian. In the phase space path integration representation, $|S_1^{'z}\rangle = e^{-\hat{H}z}|S_1^{'0}\rangle$ can be expressed as the path integration over $t^z_{k_1 k_2}$, $p^z_{k_1 k_2}$ as in Eq. (6), where the bulk action is given by $S_{bulk} = \int_0^{z^*} dz \, \mathcal{L}_{bulk}$.

The integration over $p$ in the bulk gives rise to the constraint for $t$,

$$
\partial_z t_{\tilde{k}_1,\tilde{k}_2} - 2i \int d^D \tilde{k} \, \tilde{G}(\tilde{k}) t_{\tilde{k}_1,-\tilde{k}} t_{\tilde{k}_2,\tilde{k}} + (D-2) t_{\tilde{k}_1,\tilde{k}_2} + \tilde{k}_1 (\partial_{\tilde{k}_1} t_{\tilde{k}_1,\tilde{k}_2}) + \tilde{k}_2 (\partial_{\tilde{k}_2} t_{\tilde{k}_1,\tilde{k}_2}) = 0. \tag{H.7}
$$

Treating $\{t_{\tilde{k}_1 \tilde{k}_2}\}$ as elements of matrix $\mathbf{t}$, we can write the solution as

$$
it_{\tilde{k}_1,\tilde{k}_2}(z) = \left( (i\tilde{\mathbf{t}}^0) \left[ I - \tilde{\mathbf{D}}(z)(i\tilde{\mathbf{t}}^0) \right]^{-1} \right)_{\tilde{k}_1,\tilde{k}_2}, \tag{H.8}
$$

where $\tilde{t}^0_{\tilde{k}_1,\tilde{k}_2} = e^{(2-D)z} t^0_{\tilde{k}_1 e^{-z}, \tilde{k}_2 e^{-z}}$, and $\tilde{D}_{\tilde{k},\tilde{k}'} = \tilde{D}_{\tilde{k}} \delta(\tilde{k} + \tilde{k}')$ with $\tilde{D}_{\tilde{k}} = \frac{2}{\tilde{k}^2} \left[ \exp(-\frac{\tilde{k}^2 e^{-2z}}{M^2}) - \exp(-\frac{\tilde{k}^2}{M^2}) \right]$.

We can further integrate over $p^0_{rr'}$ in $S_{UV}$ as $\int \mathcal{D}p^0_{rr'} e^{-NS_{UV}} = \prod_{r \neq r'} \delta(t^0_{rr'} - iJ_{rr'}) e^{-\frac{N}{4\lambda} \int d^D r \, (t^0_{rr} - iJ_{rr})^2}$. In the large $N$ limit, the integration over $t^0$ can be replaced with the saddle-point solution. The saddle-point equation for $t_{rr'}$ is solved if we write $-i\mathbf{t}^0 = \mathbf{J} + \mathbf{X}$, where $\mathbf{J}_{k_1,k_2} = J_{k_1} \delta(k_1 + k_2)$ is the UV hopping matrix in the momentum space, and $\mathbf{X}$ is a matrix whose elements depend only on $k_1 + k_2$ as $\mathbf{X}_{k_1,k_2} = X_{k_1+k_2}$. $\tilde{\mathbf{X}}_{\tilde{k}_1,\tilde{k}_2} \equiv e^{(2-D)z^*} \mathbf{X}_{k_1,k_2}$ satisfies

$$
\begin{aligned}
(2\pi)^D \frac{N}{2\lambda} e^{(D-4)z^*} \tilde{X}_{-\tilde{p}} &= \frac{N}{2} \int d^D \tilde{k}_1 \, \tilde{D}_{\tilde{k}_1} \left[ I + \tilde{\mathbf{D}}(\tilde{\mathbf{X}} + \tilde{\mathbf{J}}) \right]^{-1}_{\tilde{p}+\tilde{k}_1,\tilde{k}_1} \\
&+ \int d^D \tilde{k}_1 \left\{ \int d^D \tilde{k} \, \phi_{\tilde{k}} \left[ I + (\tilde{\mathbf{X}} + \tilde{\mathbf{J}}) \tilde{\mathbf{D}} \right]^{-1}_{\tilde{k},\tilde{k}_1} \right\} \left\{ \int d^D \tilde{k}' \left[ I + \tilde{\mathbf{D}}(\tilde{\mathbf{X}} + \tilde{\mathbf{J}}) \right]^{-1}_{\tilde{p}-\tilde{k}_1,\tilde{k}'} \phi_{\tilde{k}'} \right\}, \tag{H.9}
\end{aligned}
$$

where $\tilde{\mathbf{J}}_{\tilde{k}_1,\tilde{k}_2} \equiv e^{(2-D)z^*} \mathbf{J}_{k_1,k_2}$. The effective action at scale $z^*$ is written as

$$S_{tot} = -(2\pi)^D \frac{N}{4\lambda} e^{(D-4)z^*} \int d^D\tilde{p}\, \tilde{X}_{\tilde{p}} \tilde{X}_{-\tilde{p}} + \frac{N}{2} \int d^D\tilde{k}\, \log\big[I + \tilde{\mathbf{D}}(\tilde{\mathbf{X}} + \tilde{\mathbf{J}})\big]_{\tilde{k},\tilde{k}} \tag{H.10}$$
$$+ \frac{1}{2} \int d^D\tilde{k}\, G_M^{-1}(\tilde{k}) \phi_{\tilde{k}} \phi_{-\tilde{k}} + \int d^D\tilde{k}_1 d^D\tilde{k}_2 \left((\tilde{\mathbf{X}} + \tilde{\mathbf{J}})\big[I + \tilde{\mathbf{D}}(\tilde{\mathbf{X}} + \tilde{\mathbf{J}})\big]^{-1}\right)_{\tilde{k}_1,\tilde{k}_2} \phi_{\tilde{k}_1} \phi_{\tilde{k}_2}.$$

If we write $\phi$-independent part of $\tilde{\mathbf{X}}$ as $\tilde{\mathbf{x}}$, it satisfies

$$\int^{\Lambda e^{z^*}} d^D\tilde{k}_1 \tilde{D}_{\tilde{k}_1} \big[I + \tilde{\mathbf{D}}(\tilde{\mathbf{x}} + \tilde{\mathbf{J}})\big]^{-1}_{\tilde{p}+\tilde{k}_1,\tilde{k}_1} = \frac{(2\pi)^D}{\lambda} e^{(D-4)z^*} \tilde{x}_{-\tilde{p}}, \tag{H.11}$$

where $(\tilde{\mathbf{x}})_{\tilde{k}_1,\tilde{k}_2} = \tilde{x}_{\tilde{k}_1+\tilde{k}_2}$ and $\Lambda$ is the UV cutoff at $z = 0$. In the presence of the translational symmetry, $\tilde{x}_{-\tilde{p}} \propto \delta(\tilde{p})$. $\tilde{\mathbf{x}}$ can be expanded as $\tilde{\mathbf{x}} = \tilde{\mathbf{x}}_0 e^{2z^*} + \tilde{\mathbf{x}}_2 + O(e^{-2z^*})$. At the critical point, the exponentially growing part of $\tilde{\mathbf{x}}$ cancels with that of $\tilde{\mathbf{J}}$, and $\tilde{\mathbf{x}} + \tilde{\mathbf{J}} \sim \mathcal{O}(1)$. If we define $\tilde{\mathbf{X}}' \equiv \tilde{\mathbf{X}} - \tilde{\mathbf{x}}_0 e^{2z^*}$, $\tilde{\mathbf{X}}'$ satisfies

$$\frac{N}{2} \int d^D\tilde{k}_1\, \tilde{D}_{\tilde{k}_1} \big[\tilde{\mathbf{T}} + \tilde{\mathbf{D}}\tilde{\mathbf{X}}'\big]^{-1}_{\tilde{p}+\tilde{k}_1,\tilde{k}_1} - \frac{N}{2} \int d^D\tilde{k}_1\, \tilde{D}_{\tilde{k}_1} \big[\tilde{\mathbf{T}} + \tilde{\mathbf{D}}\tilde{x}_2\big]^{-1}_{\tilde{p}+\tilde{k}_1,\tilde{k}_1} \tag{H.12}$$
$$+ \int d^D\tilde{k}_1 \left\{ \int d^D\tilde{k}\, \phi_{\tilde{k}} \big[\tilde{\mathbf{T}}^T + \tilde{\mathbf{X}}'\tilde{\mathbf{D}}\big]^{-1}_{\tilde{k},\tilde{k}_1} \right\} \left\{ \int d^D\tilde{k}' \big[\tilde{\mathbf{T}} + \tilde{\mathbf{D}}\tilde{\mathbf{X}}'\big]^{-1}_{\tilde{p}-\tilde{k}_1,\tilde{k}'} \phi_{\tilde{k}'} \right\}$$
$$= (2\pi)^D \frac{N}{2\lambda} e^{(D-4)z^*} \left(\tilde{X}'_{-\tilde{p}} - \tilde{\mathbf{x}}_2\right),$$

where $\tilde{\mathbf{T}} = I + \tilde{\mathbf{D}}(\tilde{\mathbf{x}}_0 e^{2z^*} + \tilde{\mathbf{J}})$. When $D > 4$, $\tilde{\mathbf{X}}' = \mathbf{x}_2$, and the fixed point action is Gaussian. For $D < 4$, $\tilde{\mathbf{X}}'$ becomes $\phi$-dependent, and Eq. (H.10) becomes non-Gaussian.

If we deform the UV theory by adding a uniform mass $\delta J_{k_1 k_2} = \epsilon' \delta(k_1 + k_2)$, which translates to $\delta\tilde{J}_{\tilde{k}_1\tilde{k}_2} = \epsilon' e^{2z}\delta(\tilde{k}_1 + \tilde{k}_2)$, the effective action changes by

$$\Delta'_\epsilon S_{tot} \propto \epsilon' \big[e^{Dz} \tilde{x}_0 + e^{(D-2)z} \tilde{X}'_0\big]. \tag{H.13}$$

This implies that $\tilde{\mathcal{X}}'_{\tilde{r}} \equiv \int d^D\tilde{k} \tilde{X}'_{\tilde{k}} e^{i\tilde{k}\tilde{r}}$ is a local operator with with scaling dimension $\Delta_X = 2$. If a non-local hopping term is turned on between $r$ and $r'$ at UV with strength $\epsilon$, $\delta J_{k_1 k_2} = \epsilon e^{-ik_1 r - ik_2 r'}$ $(\delta\tilde{J}_{\tilde{k}_1\tilde{k}_2} = \epsilon e^{(2-D)z} e^{-i\tilde{k}_1\tilde{r} - i\tilde{k}_2\tilde{r}'})$, the change of the effective action is given by

$$\Delta_\epsilon S_{tot} \propto \epsilon e^{(2-D)z} \int d^D\tilde{k} d^D\tilde{k}'\, e^{-i(\tilde{k}\cdot\tilde{r} + \tilde{k}'\cdot\tilde{r}')} \phi_{\tilde{k}}^S \phi_{\tilde{k}'}^S, \tag{H.14}$$

in the $|\tilde{r} - \tilde{r}'| \to \infty$ limit, where $\phi_{\tilde{k}}^S = \int d^D\tilde{k}' \big[I + \tilde{\mathbf{D}}(\tilde{\mathbf{X}} + \tilde{\mathbf{J}})\big]^{-1}_{\tilde{k}',\tilde{k}} \phi_{\tilde{k}'}$. $\Phi_{\tilde{r}}^S \equiv \int d^D\tilde{k}\, e^{-i\tilde{k}\cdot\tilde{r}} \phi_{\tilde{k}}^S$ corresponds to the local scaling operator in the fundamental representation of $O(N)$ with scaling dimension $\Delta_\phi = \frac{D-2}{2}$.

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
