# Peer review of "Exact effective action for the O(N) vector model in the large N limit"

_SciPost Physics, doi:SciPost Phys. 15, 111 (2023)_

## Round 3 · Referee Report · Anonymous (Referee 1) · 2023-4-15

Report

The authors computed and solved the exact RG equations of the large N O(N) vector model, using the quantum RG technique developed by one of the authors previously. The result in this paper is novel and interesting, and is a nice demonstration of the usage of the quantum RG method.

  1. In the calculation, the authors used the large z limit which seems to me that they are looking at the RG flow that is close to a fixed point. Can the authors comment on the error in this approximation, and it what sense is the result exact?

2.Can the authors comment on the effects of 1/N corrections?

  • validity: -
  • significance: -
  • originality: -
  • clarity: -
  • formatting: -
  • grammar: -

Author:  Han Ma  on 2023-07-07  [id 3786]

(in reply to Report 1 on 2023-04-15)

We thank the referee for the suggestions. In the revised manuscript, we added one new paragraph in Sec.IV and one new section (Sec. VI) to address finite z corrections and 1/N corrections.

1)To explain the corrections that arise at a finite z, we added the followinig paragraph in page 11 :

``We note that Eq. (18) is obtained by taking the large $z^\ast$ limit of the effective action in the presence of the insertion of two UV operators. At a finite $z^\ast$, Eq. (18) is corrected by extra terms that are further suppressed by higher powers of $e^{-z^\ast}$. Since the scaling operators are eigen-operators whose forms do not change under the coarse graining and dilatation, those corrections only contribute to the sub-leading scaling operators with larger scaling dimensions. The expressions for the leading scaling operators in Eqs. (19) and (20) are exact in the large $N$ limit.

2) In Sec. VI, we include discussions on (1) how to obtain the first two 1/N corrections (in Eq.29) through an explicit computation, (2) how 1/N corrections can be obtained by studying the fluctuations of the dynamical source field in the dual bulk theory on page 14 and (3) the relation between the dynamical source field and the dynamical spacetime that emerges in the bulk.

---

## Round 3 · Referee Report · Anonymous (Referee 2) · 2023-4-24

Report

In this paper the quantum RG formalism, developed earlier by one the authors, is applied to the O(N) model, for which the Wilsonian effective action is computed at large N.
On the positive side, I think that the quantum RG is an interesting reformulation of the Wilsonian RG, and as such it can provide a fresh look at old problems.
On the negative side, the formalism and notation is not the most immediate, and at the same time, there is the impression that not much new is achieved, therefore discouraging further readings on this formalism. Therefore, it would be important to compare their results to existing literature.
As stated in the summary, and stressed by the title, the main result of the paper is equation (12), which is a closed expression for the Wilsonian effective action for the vector O(N) model in the large N limit. From the way it is presented, it would seem that this is a completely new result achieved thanks to the quantum RG formalism, but in fact it is well known that for the O(N) model in the large N limit one can compute exactly the free energy with or without sources. This can be achieved in various ways, most famously by introducing the Hubbard-Stratonovich field (e.g. Coleman, Jackiw and Politzer, Phys.Rev.D 10 (1974) 2491, or many books and reviews), but also by changing variables to bilocal fields (as in Jevicki and Sakita, NuclearPhysicsB185 (1981) 89-100, similarly to the more recent treatment of the SYK model), or by the 2PI formalism of Cornwall, Jackiw and Tomboulis, Phys.Rev.D 10 (1974) 2428-2445 (see https://inspirehep.net/literature/659724 for a good review of its application to the O(N) model at large N). All these methods apply straightforwardly also to the O(N) model with a modified free propagator, such as that with UV and IR cutoffs used to derive the exact RG equations. And indeed equation (12), with its logarithm and squared tadpole terms, looks rather familiar, although with a slightly heavier notation.
Therefore, it would be very important to compare this results with previously known effective actions, and explain what, if anything, it is gained by the quantum RG formalism.
Before seeing such a comparative discussion I cannot recommend the manuscript for publication.
  • validity: -
  • significance: -
  • originality: -
  • clarity: -
  • formatting: -
  • grammar: -

Author:  Han Ma  on 2023-07-07  [id 3785]

(in reply to Report 2 on 2023-04-24)

We thank the referee for the suggestion. It is very important to compare our results with previously known effective actions, and explain what, if anything, is gained by the quantum RG formalism. In the revised manuscript, we added the 2nd, 3rd and 4th paragraphs in the Introduction to explain (1) what is already known in the literature, (2) what is new in our paper and (3) how the quantum RG formalism is different from the earlier method based on collective fields.

Simply speaking, the three questions can be answered in the following.
(1) In the large N limit, the exact RG equation for the quantum effective potential for spacetime-independent fields was written in a closed form. The effective potential can be computed in the power series of the field. Meanwhile, at the infrared fixed point of the exact RG equation, the quantum effective action has been written down as a functional of one collective variable. However, neither the full solution of the exact RG equation nor the relation between the collective field and fundamental microscopic field is known yet.
(2) We obtain the exact solution of the RG equation in the large $N$ limit in terms of one collective variable. Such a full solution is scale dependent and its IR limit is related to the previously obtained effective action Ref[e.g. coleman1974,moshe2003] through the Legendre transformation. We are able to both identify the relation between the collective variable and the fundament field and the relation between the UV and IR operators.
(3) The quantum RG method helps us to keep track of the RG flow and find the evolution of the effective action as well as the deformation in the UV action. This allows us to obtain the new results summarized above.

---

## Round 4 · Referee Report · Anonymous · 2023-7-17

Report

The authors have partially addressed my concern by adding some comments and references in the introduction. I was rather hoping in a more explicit comparison of their equation (12) (notice that it is in overflow, which should be corrected) to other known expressions of the effective action, or at least to a more standard notation. However, this is a minor point which does not affect the content of the paper.

  • validity: -
  • significance: -
  • originality: -
  • clarity: -
  • formatting: -
  • grammar: -

Author:  Han Ma  on 2023-07-27  [id 3839]

(in reply to Report 1 on 2023-07-17)

We thank the referee for the comment. Please find the note attached, where we have shown the explicit relation between our action in Eq.(9) (and consequently Eq.(12)) and previously used effective action in terms of collective field. And then we emphasize that it is our formulation which keeps track of the RG flow and gives the relation between UV and IR scaling operators.

Simply speaking, starting from Eq.(9), we can integrate over the UV field $\phi$ and insert an auxiliary collective field $\varphi$ to get the familiar action:
$S= \int d^D r \left[ (\partial_r\varphi_r )^2+ X_r \varphi_r^2 \right]$

But the resulting action loses information in the UV. On the contrary, our effective action is always expressed in terms of the UV field and it can be used to study the relation between UV and IR scaling operators.

Attachment:

reply.pdf

---

## Round 4 · List of Changes

1. We added 2,3,4 paragraphs in the Introduction to explain (1) what is already known in the literature, (2) what is new in our paper and (3) how the quantum RG formalism is improved from the earlier method based on collective fields.
2. We added the last paragraph in Sec.IV to point out our new finding on the exact relation between the two leading scaling operators at the Wilson-Fisher fixed point.
3. We added Sec.VI to compute the 1/N corrections in the effective action.

---

## Editorial Decision

published